# Suppressing proteasome mediated processing of topoisomerase II DNA-protein complexes preserves genome integrity

Nicholas Sciascia[1,2], Wei Wu[1†], Dali Zong[1†], Yilun Sun[3], Nancy Wong[1], Sam John[1], Darawalee Wangsa[4], Thomas Ried[4], Samuel F Bunting[5], Yves Pommier[3], André Nussenzweig[1]*

[1]Laboratory of Genome Integrity, National Institutes of Health, Bethesda, United States; [2]Institute for Biomedical Sciences, George Washington University, Washington, United States; [3]Developmental Therapeutics Branch, National Institutes of Health, Bethesda, United States; [4]Genetics Branch National Cancer Institute, National Institutes of Health, Bethesda, United States; [5]Department of Molecular Biology and Biochemistry, Rutgers University, Piscataway, United States

*For correspondence:
andre_nussenzweig@nih.gov

†These authors contributed equally to this work

Competing interests: The authors declare that no competing interests exist.

**Abstract** Topoisomerase II (TOP2) relieves topological stress in DNA by introducing double-strand breaks (DSBs) via a transient, covalently linked TOP2 DNA-protein intermediate, termed TOP2 cleavage complex (TOP2cc). TOP2ccs are normally rapidly reversible, but can be stabilized by TOP2 poisons, such as the chemotherapeutic agent etoposide (ETO). TOP2 poisons have shown significant variability in their therapeutic effectiveness across different cancers for reasons that remain to be determined. One potential explanation for the differential cellular response to these drugs is in the manner by which cells process TOP2ccs. Cells are thought to remove TOP2ccs primarily by proteolytic degradation followed by DNA DSB repair. Here, we show that proteasome-mediated repair of TOP2cc is highly error-prone. Pre-treating primary splenic mouse B-cells with proteasome inhibitors prevented the proteolytic processing of trapped TOP2ccs, suppressed the DNA damage response (DDR) and completely protected cells from ETO-induced genome instability, thereby preserving cellular viability. When degradation of TOP2cc was suppressed, the TOP2 enzyme uncoupled itself from the DNA following ETO washout, in an error-free manner. This suggests a potential mechanism of developing resistance to topoisomerase poisons by ensuring rapid TOP2cc reversal.

## Introduction

Topoisomerase II (TOP2) is an evolutionarily conserved enzyme capable of generating reversible double strand breaks (DSBs) in DNA, which enables the resolution of problematic DNA topological structures that arise during normal cellular processes, such as transcription, replication, and mitosis. This process is mediated by the topoisomerase II cleavage complex (TOP2cc), a transient DNA-protein complex that is formed when each monomer of the dimeric TOP2 anchors itself to the DNA backbone via a covalent 5'-phosphotyrosyl bond. A second DNA strand can then be passed through the enzyme-bridged DSB, after which the break is re-ligated. Under physiological conditions, topoisomerases reliably execute this reaction without error, as TOP2ccs are highly reversible and short lived (*Pommier et al., 2016*). However, when they become trapped, the abortive TOP2ccs fail to re-ligate the transient DSB intermediate, presenting a severe threat to genome integrity which must be

**eLife digest** Molecules of DNA contain the archive of a cell's genetic information and identity. DNA comprises two strands that twist together into a structure known as a double helix. Physical tension tends to build up in the double helix that can cause it to break apart. To avoid this, cells have an enzyme called Topoisomerase II (TOP2) that relieves the tension by attaching itself to DNA and breaking it in a controlled way before re-sealing the break.

Drugs known as TOP2 poisons stop TOP2 from working and trap it on the DNA, which may lead to cells accumulating DNA breaks and eventually dying. Cancer cells are particularly prone to acquiring breaks in their DNA, and TOP2 poisons are therefore often used as part of chemotherapy treatments for cancer. However, it remains unclear why TOP2 poisons are more effective at killing some types of cancer cells than others.

It is thought that a molecular machine, known as the proteasome, helps cells repair the damage caused by TOP2 poisons by removing the trapped TOP2 proteins and allowing DNA repair proteins access to the broken DNA underneath. Now, Sciascia et al. have used a genetic approach to study the relationship between the proteasome and DNA repair in mouse cells exposed to TOP2 poisons.

The experiments found that when the proteasome removed TOP2 proteins that had become trapped on DNA, the subsequent DNA repair was prone to errors. Pre-treating mouse cells with another drug that inhibited the proteasome protected the cells from the effects of the TOP2 poison. Once the TOP2 poison had left the cells, the previously trapped TOP2 proteins correctly fixed the DNA and detached as they would normally. As a result, cells that had been treated with a proteasome inhibitor were more likely to survive treatment with TOP2 poisons.

Since both TOP2 poisons and proteasome inhibitors are clinically approved drugs for treating cancer they can be, and already have been, tested for use together in combination drug therapies. However, these findings suggest that caution should be taken when using these drugs together, because instead of harming the cancer cells, the proteasome inhibitors may protect the cells from the toxic effects of TOP2 poisons.

resolved quickly via the DNA repair machinery. The resolution of TOP2ccs is, therefore, crucial for cell survival.

TOP2 poisons, such as the anti-cancer agent etoposide (ETO), efficiently trap and stabilize TOP2ccs (*Muslimović et al., 2009*; *Nitiss, 2009a*; *Pommier et al., 2016*). The accumulation of these TOP2 DNA-protein complexes can create significant problems for cells by stalling replication forks and the transcription apparatus, generating torsional and genotoxic stress, which leads to the accumulation of single and double strand DNA breaks (*Cowell and Austin, 2012*; *Muslimović et al., 2009*). How these abortive TOP2ccs are recognized and targeted by the DNA damage response (DDR) remains unclear.

It has been suggested that a robust DNA damage response (DDR) is elicited only after ETO-trapped TOP2ccs have been converted to 'clean' protein-free DSB ends (*Mårtensson et al., 2003*; *Sunter et al., 2010*; *Zhang et al., 2006*). This suggests that a DSB shielded by a covalently bound protein can evade detection. Several mechanisms exist for removal of trapped TOP2ccs. However, the factors that determine how a cell chooses to deploy them are complex and remain poorly understood. The covalently-linked TOP2 protein is partially or completely degraded by the 26S proteasome, after which the residual DNA-peptide adduct is hydrolyzed by tyrosyl-DNA phosphodiesterase 2 (TDP2) to release the remnants of the entrapped protein (*Ledesma et al., 2009*; *Gómez-Herreros et al., 2013*; *Lee et al., 2018*; *Sunter et al., 2010*; *Zagnoli-Vieiral and Caldecott, 2017*). Current evidence suggests that transcription increases the rate at which TOP2 is processed by the proteasome (*Canela et al., 2019*). It has also been reported that TDP2, with the co-operation of secondary cofactors, can resolve unprocessed TOP2ccs in a proteasome-independent manner (*Schellenberg et al., 2017*).

An alternate, more error-prone mechanism for TOP2cc removal is mediated by MRE11 endonucleolytic cleavage in the vicinity of a trapped TOP2cc, which releases the entire TOP2 DNA-protein complex that results in the loss of small stretches of DNA (*Hoa et al., 2016*; *Lee et al., 2012*). Either of these mechanisms are able to produce protein-free DNA breaks that can then be recognized and

repaired by the major cellular DSB repair pathways: non-homologous end joining (NHEJ) and homologous recombination (HR).

It has long been recognized that trapped TOP2ccs are intrinsically reversible upon ETO washout (*Hsiang and Liu, 1989*; *Long et al., 1985*). Consistently, we observed that inhibiting the proteasome prior to ETO treatment enhanced the number of rapidly reversible ETO-stabilized TOP2ccs across the genome (*Canela et al., 2019*). Other studies have shown that inhibiting the proteasome not only preserves the reversibility of TOP2ccs but also is able to suppress DDR signaling (*Mao et al., 2001*; *Zhang et al., 2006*). However, the implications of suppressing the DDR in response to ETO using proteasome inhibitors, on both the long-term genome integrity and the overall viability of the cell, have not been fully explored. Considering that both proteasome inhibitors and topoisomerase poisons are used, sometimes in tandem, as frontline chemotherapeutic agents to treat a variety of cancers (*Cowell and Austin, 2012*; *Dittus et al., 2018*; *Manasanch and Orlowski, 2017*; *Thomas et al., 2017*), understanding how TOP2cc reversibility may impact the effectiveness of these drugs is clinically relevant.

Here, we utilize high-resolution genome-wide mapping of DSBs by END-seq (*Canela et al., 2016*) to examine the impact of proteasome inhibition on the fate of TOP2ccs. We found that once ETO-stabilized TOP2ccs had been proteolytically processed throughout the genome, and a robust DDR had been initiated, the large number of newly generated protein-free DSBs were repaired in a highly error-prone manner, resulting in toxic chromosomal translocations and rearrangements that led to cell death. Notably, the major DSB repair pathways (NHEJ and HR) contribute to the mis-repair of ETO-induced DSBs following proteasomal processing. By inhibiting proteasome-mediated degradation of TOP2ccs either through chemical inhibition or by ablation of RNF4-mediated ubiquitination of TOP2ccs, an intact and enzymatically competent TOP2 was able to re-seal the protein-linked DSBs without invoking a significant DDR. As a result, cells were protected from genomic instability, which ultimately led to enhanced cell survival after treatment with topoisomerase poisons.

## Results

### Proteasomal activity is necessary for triggering a DNA damage response following ETO treatment

Building on previous observations that proteasome inhibition can specifically attenuate the induction of γ-H2AX by ETO (*Mao et al., 2001*; *Zhang et al., 2006*), we sought to first confirm these findings in mouse primary splenic B-cells. Since ETO stabilizes both TOP2A and TOP2B isoforms, resting B-cells were activated with cytokines for 12 hr (*Figure 1A*). This short-term treatment ensures that B-cells remain in G1 where only the TOP2B isoform is expressed (*Canela et al., 2017*). This treatment regime, therefore, allows us to minimize any differential effect of ETO on each isoform (*Errington et al., 2004*; *Willmore et al., 1998*) and to bypass essential TOP2-dependent processes, such as DNA replication.

In agreement with other studies (*Mao et al., 2001*; *Zhang et al., 2006*), we found that treatment with a high (50 µM) dose of ETO for 2 hr elicited a strong γ-H2AX response in B-cells (*Figure 1B*). Pre-incubating cells with proteasome inhibitors (10 µM MG132, 10 µM Bortezomib (BTZ), 2 µM Epoxomicin (EPN), or 10 µM Ixazomib (IXA)) for 1 hr prior to ETO treatment almost completely abolished the γ-H2AX signal (*Figure 1B*). The suppressed γ-H2AX signal occurred specifically in response to ETO, as pre-treating cells with the same dose of proteasome inhibitor did not affect the γ-H2AX response to ionizing γ-irradiation (5 Gy IR) (*Figure 1B and C*). Thus, proteasomal activity is required for triggering a robust DDR specifically in response to ETO treatment.

### Proteasome inhibition promotes the accumulation of reversible TOP2ccs

As trapped TOP2ccs are subjected to proteasomal degradation, unprocessed TOP2ccs accumulate on DNA in cells pre-treated with a proteasome inhibitor compared to cells treated with ETO alone (*Lee et al., 2016*). To confirm that proteasome inhibitors were having the same effect in our experimental system, we quantified TOP2ccs using the ICE assay (*Anand et al., 2018*) in ETO-treated primary B-cells pre-incubated with or without BTZ. In the presence of ETO, TOP2ccs were readily detected on DNA (*Figure 1D*). Immediately after ETO was washed out, however, the amount of

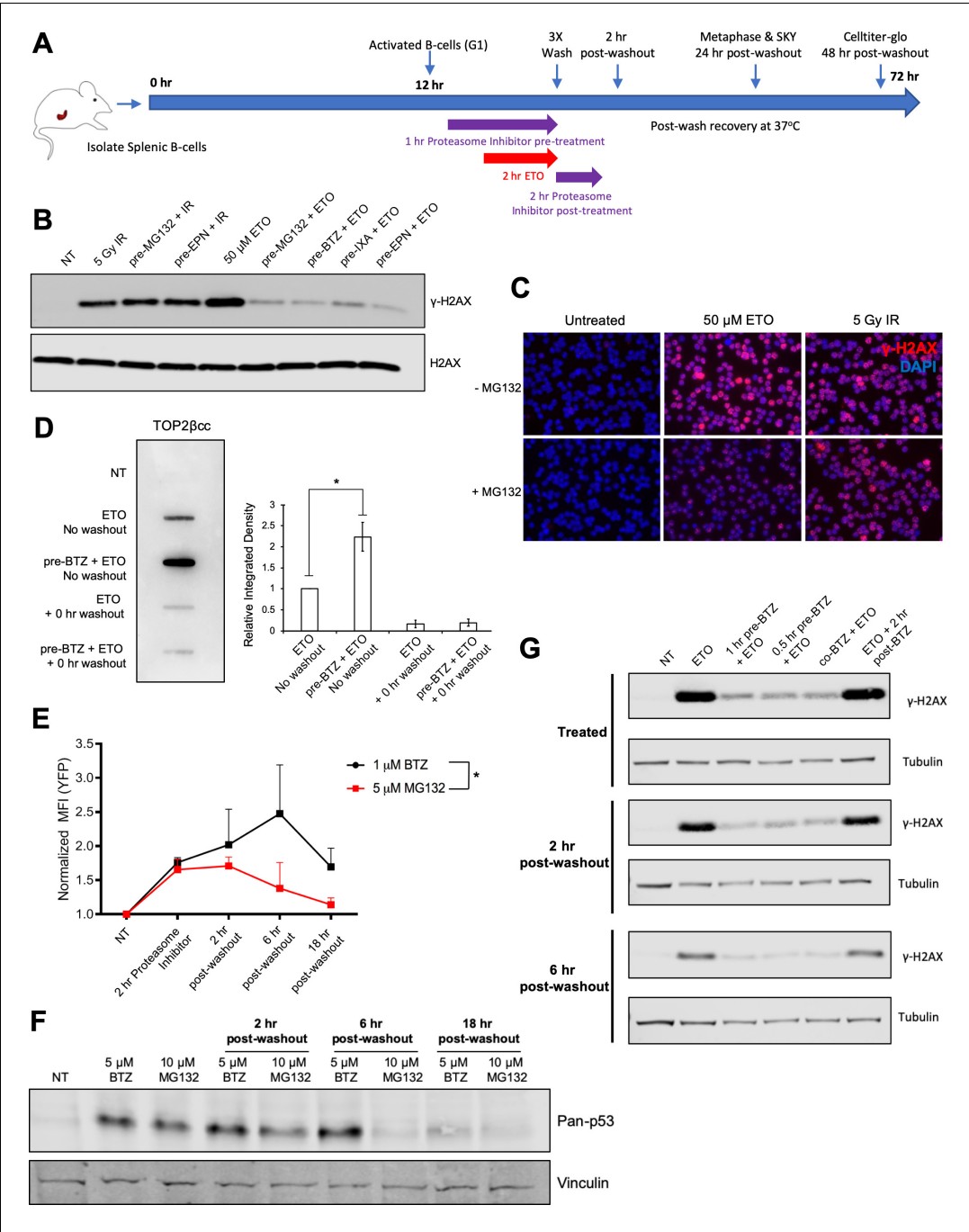

**Figure 1.** Proteasome machinery is essential for triggering DNA damage response to ETO treatment. (**A**) As indicated in the schematic, primary splenic B-cells were isolated from mouse spleens and activated with a cocktail of Il-4, LPS, and RP105. 12 hr post-activation, while the B-cells were still in the G1 phase of the cell cycle, they were pre-treated with proteasome inhibitor (10 µM MG132 or 5 µM Bortezomib) for 1 hr prior to an additional 2 hr co-treatment with 50 µM ETO. Following the ETO pulse, the cells were washed with ice cold drug-free media (3 × 5 min spin at 1500 rpm and 4°C to pellet B-cells between washes; 15–20 min total time to complete washout), then returned to fresh drug free media and allowed to recover at 37°C for up to 48 hr. In the case of post-treatment, proteasome inhibitor was added to the wash media and then the cells were incubated for an additional 2 hr at 37°C with proteasome inhibitor only. For western blot and ICE assay analysis, cells were harvested immediately following drug treatment or washout (0 hr washout). For metaphase and SKY analysis, cells were harvested 24 hr following washout. For CellTiter-Glo viability, cells were harvested 48 hr following washout. For END-seq analysis, cells were harvested before drug washout, immediately following washout (0 hr washout), or after a 2 hr recovery

*Figure 1 continued on next page*

*Figure 1 continued*

in drug free media (2 hr washout). (B) γ-H2AX western blot in G1 WT primary splenic B-cells following 2 hr 50 μM ETO treatment ±1 hr proteasome inhibitor pre-treatment or 30 min following 5Gy IR ±1 hr proteasome inhibitor pre-treatment. Proteasome inhibitors tested: 10 μM MG132, 10 μM BTZ (Bortezomib), 2 μM EPN (Epoxomicin), 10 μM IXA (Ixazomib). (C) γ-H2AX immunofluorescence staining (red) in G1 arrested WT pre B cells following 2 hr, 50 μM ETO treatment ±1 hr 10 μM MG132 pre-treatment or 30 min after 5 Gy IR treatment ±1 hr 10 μM MG132 pre-treatment. (D) WT primary splenic B-cells (N = 3) in G1 were treated for 2 hr with 50 μM ETO ±1 hr, 5 μM BTZ pre-treatment. DNA was then isolated from cells following the ICE assay protocol and probed with anti-TOP2β antibody to quantify levels of TOP2βcc. Relative band intensity was measured for each sample and averaged for all three mice (ETO vs. pre-BTZ + ETO p=0.0168; ETO vs. ETO + 0 hr washout p=0.0045; pre-BTZ + ETO vs. pre-BTZ + ETO 0 hr washout p=0.0046; statistical significance calculated using student T-test). (E) eHAP cells were transfected for 48 hr with a YFP-tagged degron construct, which is degraded in a proteasome-dependent manner (*Bence et al., 2001*). Cells were then treated with 5 μM MG132 or 1 μM BTZ for 2 hr and then the drugs were washed out (3X wash in cold drug free media). Cells were fixed before washout with drugs still present or at 2 hr, 6 hr, and 18 hr post-washout. YFP signal intensity was quantified by FACS. (BTZ (before washout) vs. NT p=0.025; 2 hr post-washout MG132 vs NT p=0.049; statistical significance calculated using student T-test. MG132 vs BTZ p=0.038; statistical significance calculated using two column ANOVA). (F) pan-p53 levels in primary splenic B-cells treated for 3 hr with 10 μM MG132 or 5 μM BTZ. Drugs were then washed out and samples collected before washout or at 2 hr, 6 hr and 18 hr post-washout. (G) γ-H2AX levels in primary splenic B-cells as determined by Western blotting. Cells were treated with 50 μM ETO for 2 hr ±1 hr or 30 min pre-treatment, co-treatment and post-treatment with 5 μM BTZ at three different timepoints: before washout (treated), 2 hr and 6 hr post-washout.

DNA-bound TOP2 diminished strongly (*Figure 1D*). The proteasome continuously degraded ETO-trapped TOP2ccs, as evidenced by the markedly enhanced accumulation of TOP2ccs when cells were pre-incubated with BTZ (*Figure 1D*). The majority of these TOP2ccs also dissociated from DNA immediately following ETO washout (*Figure 1D*). Thus, inhibiting proteasomal degradation appears to preserve the integrity of TOP2ccs enabling reversal of complexes by completion of the enzymes' catalytic cycle upon drug removal.

Alternatively, proteasome activity could recover immediately after the removal of ETO and the proteasome inhibitor and would similarly lead to a loss of TOP2cc signal in the ICE assay. To determine the duration and reversibility of proteasomal inhibition, we directly monitored proteasome activity in eHAP cells transfected with a YFP-Degron reporter that has been demonstrated to be a *bona fide* proteasomal substrate (*Bence et al., 2001*). As expected, a 2 hr treatment with MG132 or BTZ significantly increased the YFP signal from baseline (*Figure 1E*). Following washout of MG132 or BTZ, the elevated YFP signal persisted for several hours before it began to decrease and BTZ appeared to be a more potent and persistent inhibitor of the proteasome than MG132 (*Kisselev and Goldberg, 2001*). As an additional measure of proteasome activity, we quantified the protein levels of p53, as it is known to be stabilized upon proteasome inhibition (*An et al., 2000*; *Halasi et al., 2014*). Consistent with the YFP-degron results in eHAP cells, we observed that p53 protein remained stabilized in primary B-cells for several hours after proteasome inhibitors were washed out, with BTZ again being more potent than MG132 (*Figure 1F*).

Thus, proteasome activity is not readily recovered even after the removal of proteasome inhibitor, suggesting that the rapid loss of ETO-induced TOP2ccs in MG132 pre-treated cells upon washout most likely reflects the reversal of TOP2ccs by completion of the enzymes' catalytic cycle upon drug removal. Accordingly, we did not observe a delayed γ-H2AX induction at either 2 hr or 6 hr post-ETO and BTZ washout, suggesting that proteasomal activity remain suppressed for at least several hours post-washout (*Figure 1G*). These data imply that persistent proteasome inhibition allows for TOP2cc reversal and prevents trapped TOP2ccs from being converted into protein-free DSB ends that are capable of eliciting a robust DNA damage response (DDR).

## Timing of proteasome inhibition determines its impact on TOP2 metabolism

Contrary to our observations, previous studies have shown that proteasome inhibitors synergize with topoisomerase poisons like ETO in mediating cell killing (*Aras and Yerlikaya, 2016*; *Ceruti et al., 2006*; *Destanovic et al., 2018*; *Dittus et al., 2018*; *Lee et al., 2016*; *von Metzler et al., 2009*). Interestingly, we found that the addition of BTZ prior to or concurrent with ETO suppressed DDR

signaling, but incubating B-cells with BTZ post-ETO treatment did not (*Figure 1G*). These results showed that the timing of proteasome inhibitor treatment relative to ETO treatment is critical to its effects on TOP2cc metabolism and subsequent DDR signaling.

## Proteasome inhibition decreases the persistence of ETO-induced TOP2-mediated DSBs by promoting TOP2cc reversibility

To further analyze the influence of the proteasome on TOP2ccs, we employed genome-wide DSB mapping by END-seq (*Canela et al., 2019*; *Canela et al., 2016*). While ETO can generate high levels of both SSBs and DSBs (*Baranello et al., 2014*; *Gittens et al., 2019*), END-seq only detects DSBs generated by ETO (*Canela et al., 2017*; *Canela et al., 2016*). However, this protocol allows us to capture and distinguish both TOP2ccs and protein-free DSBs generated by ETO (*Canela et al., 2019*). First, we assessed whether proteasome inhibition blocked TOP2 from making incisions in DNA. To this end, we used a cocktail of Exonuclease VII (ExoVII) and Exonuclease T (ExoT) during sample preparation, allowing us to capture all sites of TOP2 activity (*Canela et al., 2019*). As expected, END-seq analysis revealed that TOP2-mediated DSBs were readily detectable in ETO-treated B-cells (*Figure 2A–C*). The number and intensity of TOP2-mediated DSBs were reproducible across experiments (*Figure 2—figure supplement 1A–B*). Consistent with the results shown in *Figure 1D*, we found that BTZ pre-treatment did not dramatically affect the overall number or location of TOP2-mediated DSBs (*Figure 2A–C*; *Figure 2—figure supplement 1C–D*); TOP2-mediated DSBs were still strongly induced in the BTZ pre-treated sample, despite eliciting only a weak γ-H2AX response (*Figure 1B*). Thus, while proteasome inhibition does not affect the ability of TOP2 to cleave DNA, it helps maintain the resultant DSBs in a protein-linked form (persistent TOP2cc) that conceals them from the DDR. Nevertheless, we did observe a small decrease in the overall intensity of the TOP2-mediated DSB signal (*Figure 2A and C*) upon BTZ pre-treatment. This could be due to reversal of TOP2ccs in the BTZ pre-treated sample by completion of the enzymes' catalytic cycle upon drug removal.

Since inhibiting proteasomal degradation did not affect the initial formation of TOP2-mediated DSBs, we wanted to understand if proteasome inhibition affected their fate over time. To directly evaluate the impact of proteasome inhibition on repair of TOP2-mediated DSBs, we used END-seq to assess their longevity by harvesting primary B-cells both immediately following ETO/BTZ washout (0 hr washout) and after a 2 hr recovery in drug-free media at 37°C (2 hr washout). Notably, BTZ exerted a substantial impact on the rate at which TOP2-mediated DSBs resolved over the course of the 2 hr washout, with very few detectable breaks remaining after recovery relative to cells treated with ETO alone (*Figure 2A and D*; *Figure 2—figure supplement 1E*). We plotted the fraction of TOP2-mediated DSBs at each washout timepoint relative to the initial level of breakage following ETO treatment (*Figure 2D*). Using this analysis, we found that immediately following drug washout, 30% of the initial TOP2-mediated DSBs were still detectable in cells treated with ETO only, while just 15% were still present in the corresponding BTZ pre-treated sample (*Figure 2A and D*). After a 2 hr recovery, B-cells treated with ETO alone still had a detectable break signal (17%), whereas in BTZ-pretreated cells, it returned almost to baseline (4%, *Figure 2A and D*). This suggested that inhibiting the proteasome enhanced the amount of TOP2ccs which rapidly reverse following ETO washout.

As a second form of analysis, we normalized the TOP2-mediated DSB intensity at each post-washout timepoint. We introduced a new parameter, termed TOP2-mediated DSB persistence, which was calculated as the ratio of the peak intensity (RPKM) for individual TOP2-mediated DSBs at each timepoint (either 0 hr washout or 2 hr washout) relative to its initial intensity in the corresponding pre-washout sample (*Figure 2E*; *Figure 2—figure supplement 1F*). Through this analysis, we found that approximately 20% of the initial TOP2-mediated DSBs were classified as persistent immediately following ETO washout (0 hr washout), and 13% of initial TOP2-mediated DSBs were still persistent 2 hr post-washout (2 hr washout) (*Figure 2E*). By contrast, BTZ pre-treatment more significantly reduced the fraction of persistent TOP2-mediated DSBs both immediately following washout (~13%), as well as at 2 hr post-washout (~3.5%) (*Figure 2E*). These results, in conjunction with the ICE assay (*Figure 1D*), suggested that inhibiting the proteasome enhances TOP2cc reversal following ETO washout.

To directly measure the reversibility of TOP2ccs, we took advantage of the fact that ExoT lacks the ability to process protein-linked DSBs (persistent TOP2ccs that still have an intact 5'-

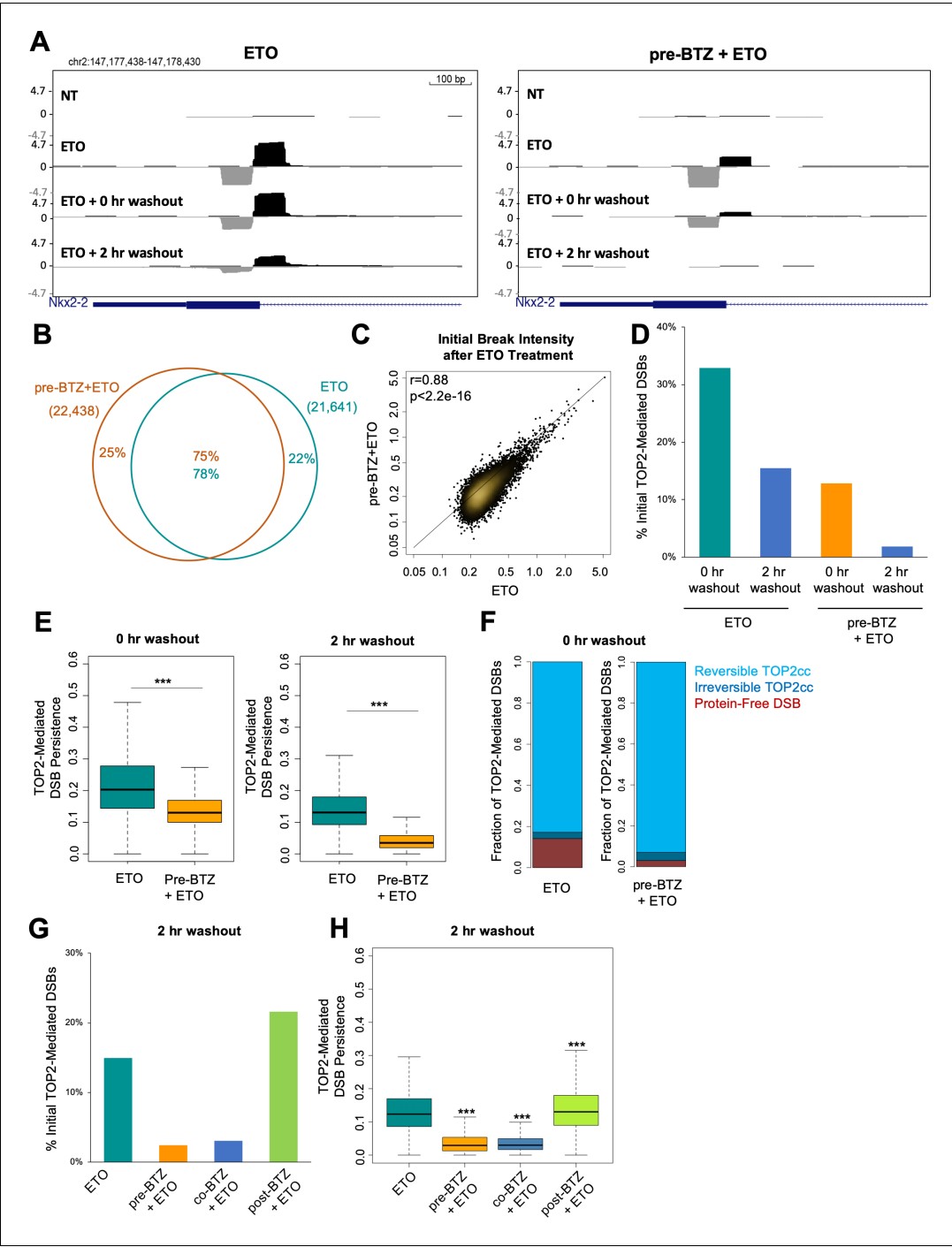

**Figure 2.** Proteasome inhibition decreases TOP2-mediated DSB persistence following ETO washout. (**A**) UCSC genome browser snapshot of a TOP2-mediated DSB captured within the Nkx2-2 gene locus by END-seq (ExoVII +ExoT) at three timepoints: in primary splenic B-cells treated for 2 hr with 50 µM ETO ± 1 hr 5 µM BTZ pre-treatment, immediately following drug washout (0 hr washout), and at 2 hr post-washout (2 hr washout). (**B**) Venn diagram depicting the overlap between TOP2-mediated DSBs generated by ETO genome-wide in the 50 µM ETO only sample (green) and the 5 µM BTZ pre-treatment sample (orange). WT primary splenic B-cells were treated for 2 hr with 50 µM ETO ± 1 hr 5 µM BTZ pre-treatment and processed by END-seq using ExoVII+ExoT. (**C**) Scatterplot depicting initial intensity of TOP2-mediated DSBs induced by ETO genome-wide in the 50 µM ETO only sample vs. the 5 µM BTZ pre-treatment sample (ETO vs. pre-BTZ + ETO p<2.2e$^{-16}$; statistical significance calculated by Pearson's correlation coefficient). WT primary splenic B-cells were treated for 2 hr with 50 µM ETO ± 1 hr 5 µM BTZ pre-treatment. (**D**) Bar graph measuring the number of TOP2-mediated DSBs (END-seq with

*Figure 2 continued on next page*

*Figure 2 continued*

ExoVII+ExoT) at 0 hr and 2 hr post-washout timepoints, represented as a fraction of the initial number of TOP2-mediated DSBs generated genome-wide (% initial breaks after ETO treatment determined by peak calling). WT primary splenic B-cells were treated for 2 hr with 50 μM ETO treatment ±1 hr 5 μM BTZ pre-treatment. (E) Box-plot of TOP2-mediated DSB persistence (relative intensity of initial breakage) genome-wide at 0 hr and 2 hr post-washout timepoints (left and right panels respectively). Green box 50 μM ETO only sample and orange box 5 μM BTZ pre-treatment sample (ETO vs. pre-BTZ + ETO $p<2.2e^{-16}$, statistical significance calculated by student T-test). WT primary splenic B-cells were treated for 2 hr with 50 μM ETO treatment ±1 hr 5 μM BTZ pre-treatment prior to washout. (F) Bar graph depicting the percentages of total TOP2-mediated DSBs (measured by ExoVII+ExoT), protein-free DSBs (measured by ExoT), and TOP2ccs (inferred by subtracting protein free DSBs from the levels of total TOP2-mediated DSBs). Reversible TOP2ccs (light blue), irreversible TOP2ccs (dark blue), and protein-free DSBs (red). WT primary splenic B-cells were treated for 2 hr with 50 μM ETO treatment ±1 hr 5 μM BTZ pre-treatment. (G) Bar graph depicting the number of TOP2-mediated DSBs at 2 hr post-washout (END-seq with ExoVII+ExoT), represented as a fraction of the initial number of TOP2-mediated DSBs measured genome-wide (% initial breaks after ETO treatment determined by peak calling). WT primary splenic B-cells were treated for 2 hr with 50 μM ETO treatment ±pre treatment, co-treatment, and post-treatment with 5 μM BTZ. Post treatment with BTZ was for 2 hr. (H) Box-plot depicting TOP2-mediated DSB persistence genome-wide at 2 hr post-washout ($p<2.2e^{-16}$ for ETO vs. pre-BTZ + ETO, ETO vs co-BTZ + ETO, and ETO vs post-BTZ + ETO; statistical significance calculated by student T-test). WT primary splenic B-cells were treated for 2 hr with 50 μM ETO treatment ±pre treatment, co-treatment, and post-treatment with 5 μM BTZ.

The online version of this article includes the following figure supplement(s) for figure 2:

**Figure supplement 1.** TOP2-Medated DSBs are reproducibly captured by END-seq.

phospotyrosyl bond) and therefore allows for the detection of only protein-free (i.e. proteolytically processed) DSBs (*Canela et al., 2019*). As such, the levels of TOP2ccs can be estimated by the difference between total TOP2-mediated DSBs detected by ExoVII+ExoT and protein-free DSBs detected by ExoT alone (*Canela et al., 2019*). Our analyses revealed that 76% of TOP2-mediated DSBs reversed immediately following washout (0 hr washout), with the majority of remaining lesions being converted into protein-free DSBs (*Figure 2F*). The fraction of reversible TOP2ccs was even higher in the BTZ pre-treated cells, as 90% of the initial TOP2-mediated DSBs were reversible immediately following washout (*Figure 2F*). Thus, suppressing proteasomal degradation of TOP2ccs led to increased TOP2cc self-reversal.

Finally, we assessed how the timing of proteasome inhibition impacted the rate at which TOP2-mediated DSBs are resolved. We found that compared to ETO alone, both BTZ pre-treatment and co-treatment significantly reduced the number and persistence of TOP2-mediated DSBs after a 2 hr washout (*Figure 2G and H*). By contrast, BTZ post-treatment slightly increased the number, as well as persistence, of TOP2-mediated DSBs at the same timepoint (*Figure 2G and H*). These results confirmed that the timing of proteasome inhibition relative to ETO treatment determined its impact on the resolution of TOP2-mediated DSBs.

## Inhibiting proteolytic degradation of TOP2ccs suppresses DNA end resection of TOP2-mediated DSBs

Since protein-free DSBs persist for a significant period of time following ETO washout, it enabled the study of DNA processing at these sites. We observed clear evidence of 5' to 3' DNA end-resection of TOP2-mediated DSBs immediately after drug washout (*Figure 3A*). After 2 hr post ETO treatment, the extent and intensity of resection had increased (*Figure 3A*), indicating ongoing end-processing at persistent TOP2-mediated DSBs. In ETO-treated cells pre-incubated with BTZ, while there were still low levels of protein-free DSBs present immediately following drug washout (0 hr washout), we could barely detect end-resection at individual breaks (*Figure 3A*). Genome-wide, we were able to identify 1289 resected breaks in ETO-treated cells immediately following washout, while only 496 resected breaks were detected in BTZ pre-treated cells, a 2.5-fold decrease (*Figure 3B*). The end-resection detected in ETO-treated cells genome-wide was reproducible across replicates (*Figure 3—figure supplement 1*). Therefore, inhibiting the proteolytic degradation of TOP2ccs prevented DNA repair-associated nucleases from processing TOP2-mediated DSBs. Moreover, the extent of resection, defined as the maximum distance away from the break summit, was significantly shorter in BTZ-pretreated cells compared to cells treated with ETO alone (*Figure 3C*;

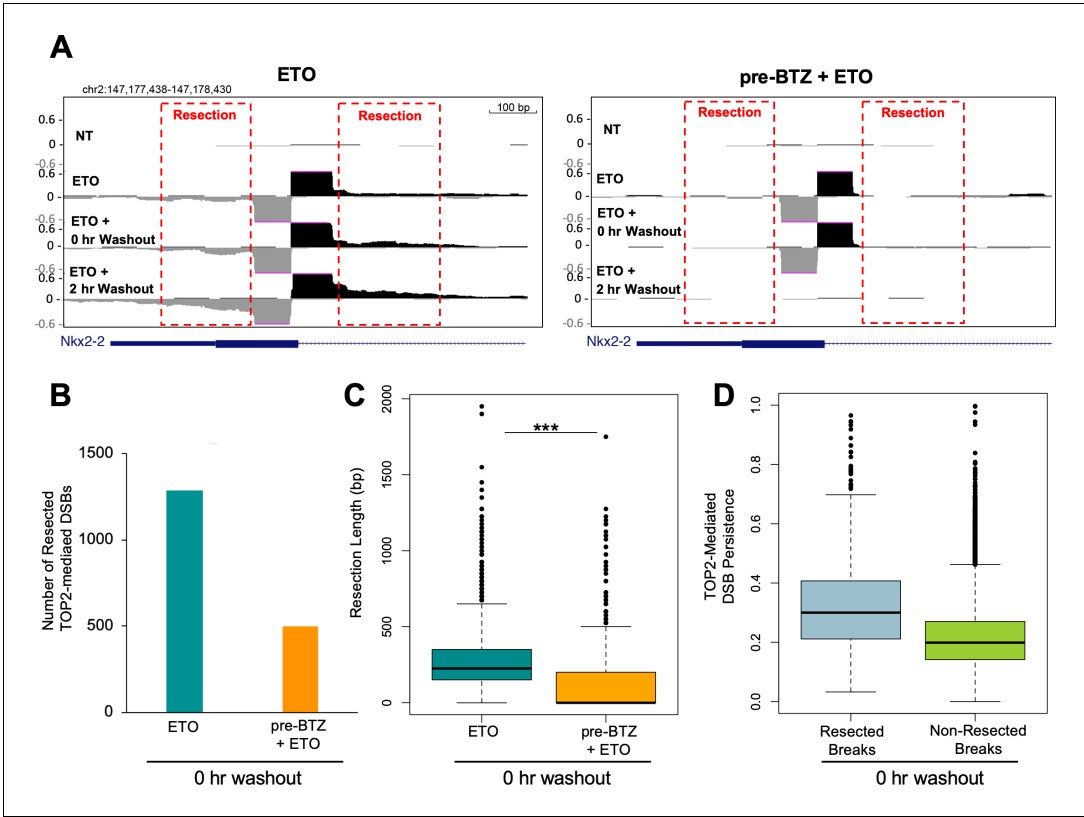

**Figure 3.** Proteasomal degradation of the TOP2cc is associated with DSB resection. (**A**) Zoomed in UCSC genome browser snapshot of the same TOP2-mediated DSB as in *Figure 2A* at three timepoints: 2 hr with 50 μM ETO (ETO), immediately following drug washout (0 hr WO), and at 2 hr post-washout (2 hr WO) (left panel). In right panel, WT primary splenic B-cells were pre-treated for 1 hr with 5 μM BTZ pre-treatment prior to washout. Red box indicates areas with resection signal. (**B**) Bar graph depicting the number of TOP2-mediated DSBs that had a resection signal immediately following drug washout (0 hr washout) in WT primary splenic B-cells treated for 2 hr with 50 μM ETO ±5 μM BTZ pre-treatment. Green bar- 50 μM ETO only sample and orange bar- 5 μM BTZ pre-treatment sample. (**C**) Box-plot quantifying maximum resection distance for all persistent TOP2-mediated DSBs genome-wide immediately following drug washout (0 hr washout) in WT primary splenic B-cells treated for 2 hr with 50 μM ETO ±5 μM BTZ pre-treatment. Green box 50 μM ETO only sample and orange box 5 μM BTZ pre-treatment sample (ETO vs. pre-BTZ + ETO $p<2.2e^{-16}$; statistical significance calculated by student T-test). (**D**) Box-plot of resected (grey) and non-resected (green) TOP2-mediated DSBs genome-wide immediately following drug washout (0 hr washout) in WT primary splenic B-cells treated for 2 hr with 50 μM ETO ± 5 μM BTZ pre-treatment. TOP2-mediated DSB resection values are plotted as a function of their persistence at the 0 hr washout timepoint. The online version of this article includes the following figure supplement(s) for figure 3:

**Figure supplement 1.** TOP2-Medated DSB resection is reproducibly captured by END-seq.

*Figure 3—figure supplement 1B*). Finally, we found that after ETO removal, TOP2-mediated DSBs undergoing nucleolytic processing (resected breaks) tended to be more persistent than non-resected breaks (*Figure 3D*; *Figure 3—figure supplement 1C*). Thus, the longer a TOP2-mediated DSB persists in the genome, the more likely it will be processed and repaired in a potentially error-prone manner.

## Proteasome inhibitor pre-treatment prevents ETO-induced genome instability and cytotoxicity

Enzymatic reversal of TOP2ccs is predicted to be an error-free process. By contrast, the degradation of TOP2ccs followed by DSB repair is potentially more susceptible to error. Nevertheless, repair of ETO-induced DSBs by TDP2-dependent NHEJ might be performed with fidelity (*Gómez-Herreros et al., 2013*) since it involves simple re-ligation of compatible 4 bp overhangs. To directly

address this question, we exposed primary B-cells in G1 to ETO (2 hr, 50 uM) with or without BTZ pre-treatment, or post-treatment. After washing out the drugs, cells were allowed to recover at 37°C in fresh, drug-free media for 24 hr before they were harvested for mitotic chromosome analysis (*Figure 1A*). Notably, we found that even a short pulse of ETO induced a substantial number and variety of chromosomal aberrations in WT B-cells (~14 aberrations per cell) (*Figure 4A and B*). The majority of aberrations that we observed were dicentric chromosomal fusions and chromosome breaks (*Figure 4A and B*). Post-treatment with MG132 did not mitigate the genotoxic effects of ETO (*Figure 4B*), consistent with the lack of effect it had on ETO-induced γ-H2AX response (*Figure 1G*). However, pre-treating B-cells with MG132 or BTZ completely protected cells from accruing chromosomal aberrations induced by ETO, with less than one aberration detected per cell (*Figure 4A and B*). Therefore, the repair of DSBs that arise from proteasome-mediated degradation of TOP2ccs is error-prone, while preventing TOP2cc processing preserves genome integrity.

Mitotic chromosome analysis is limited in its ability to determine the extent of chromosomal fusion events, as pieces of two or more chromosomes could be ligated together and still appear as a normal chromosome when stained with only DAPI. We consistently detected the presence of elongated chromosomes in mitotic spreads prepared from ETO-treated B-cells, suggesting that these might result from complex fusions involving multiple chromosomal fragments. To further characterize these seemingly intact long chromosomes, we performed spectral karyotyping (SKY) (*Liyanage et al., 1996*). As shown in *Figure 4C and D*, SKY analysis revealed that ETO-induced chromosomal fusions were extensive and complex. Indeed, these events resembled chromothripsis, in which fragments originating from three or more chromosomes are fused together (*Zhang et al., 2013*). Strikingly, pre-treating B-cells with MG132 before ETO completely protected them from all types of chromosomal fusions (*Figure 4C and D*). These results highlight the severe genotoxic consequences of a transient high dose pulse of ETO and demonstrate that such ETO-induced genome instability is proteasome-dependent.

Consistent with the impact of the proteasome on genome integrity, we found that pre-treatment with MG132, but not post-treatment, mitigated the induction of cell cycle arrest by ETO evidenced by the increased cellular incorporation of EdU after the start of ETO treatment (*Figure 4E*). While proteasome inhibition itself caused a reversible block on S-phase entry, as previously described (*Rastogi and Mishra, 2012*), proteasome inhibitor pre-treatment but not post-treatment, significantly diminished the cytotoxicity of ETO (*Figure 4F*). Taken together, our data indicate that proteasome inhibitor pre-treatment maintains cellular proliferative capacity and long-term viability by blocking ETO-induced genome instability.

## Misrepair of DSBs following TOP2 degradation is mediated by multiple DNA repair pathways

To determine if DDR signaling per se is responsible for the error-prone repair of TOP2-associated DSBs, we conducted mitotic spread analyses in ATM[-/-] and H2AX[-/-] primary B-cells. We found that these DDR signaling deficient mutants accumulated somewhat higher levels of chromosomal aberrations following ETO treatment compared to WT (*Figure 5A*). Importantly, proteasome inhibitor pre-treatment almost completely abolished ETO-induced damage in WT, as well as in ATM[-/-] and H2AX[-/-] B-cells (*Figure 5A*). Thus, proteasome inhibition but not ATM-γ-H2AX driven DDR signaling protects against ETO-induced genome instability.

Both NHEJ and HR are thought to contribute to the repair of ETO-induced DSBs (*Ledesma et al., 2009*; *Gómez-Herreros et al., 2013*; *Gómez-Herreros et al., 2017*; *Hoa et al., 2016*; *Pommier et al., 2016*). To determine whether classical NHEJ or HR is primarily responsible for the misrepair of DSBs, we tested primary B-cells deficient in the key end joining factor DNA ligase IV (Lig4[-/-]) or the key HR factors BRCA1 (BRCA1[Δ11]) and BRCA2 (BRCA2[-/-]). To this end, primary resting (G0) Lig4[-/-], BRCA1[Δ11] and BRCA2[-/-] B-cells were first activated with cytokines for 12 hr and 24 hr to drive G1- or S-phase entry, respectively, at which point they were exposed to ETO. After ETO was washed out, B-cells were given an additional 24 hr to recover before mitotic spread analysis. We found that loss of Lig4, BRCA1 or BRCA2 failed to mitigate aberrant chromosomal rearrangements in ETO-treated G1 and cycling B-cells, respectively (*Figure 5B and C*; *Figure 5—figure supplement 1*). Furthermore, MG132 pre-treatment completely prevented ETO-induced genome instability in Lig4[-/-] and BRCA1[Δ11] cells (*Figure 5B and C*). Thus, the protective effects of

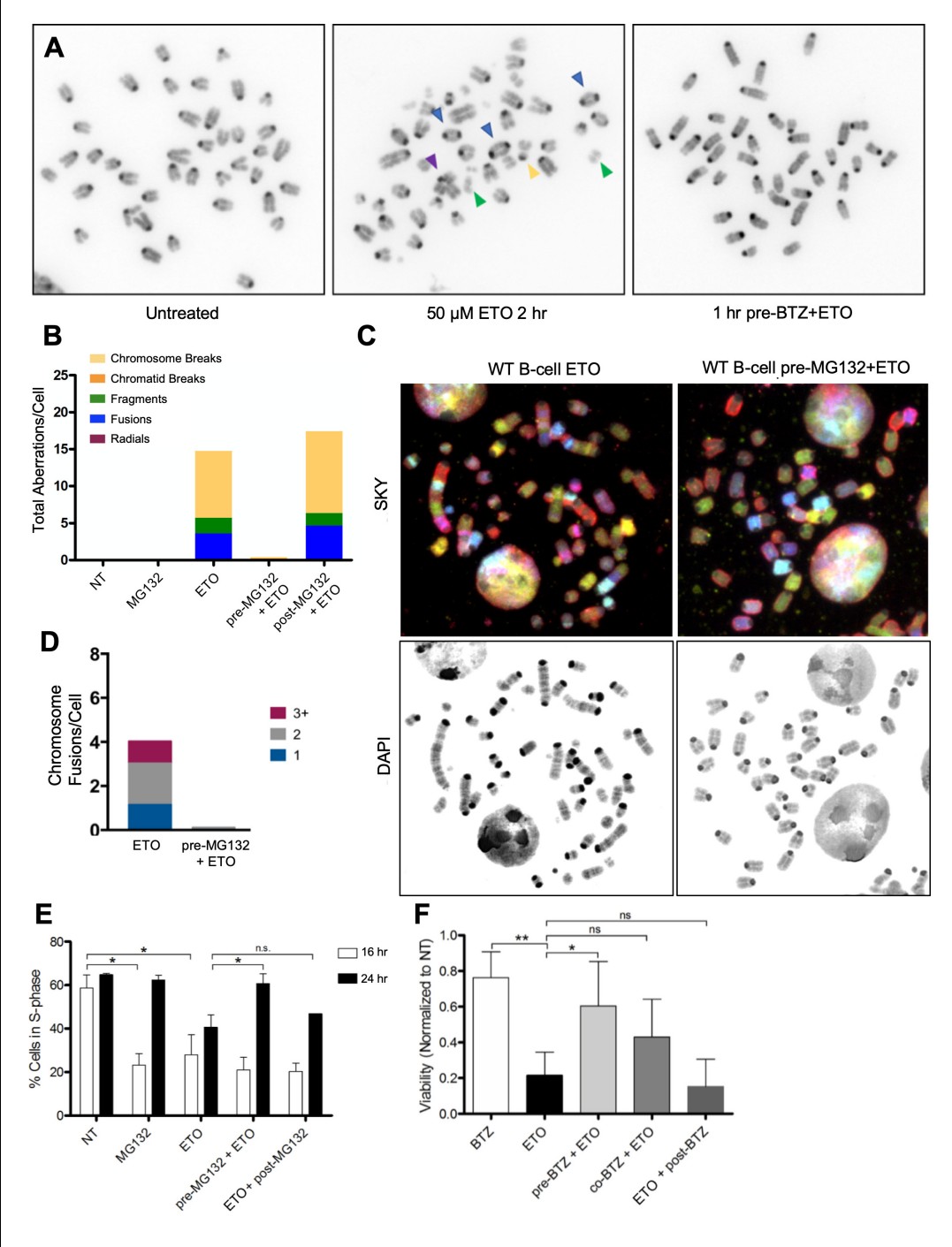

**Figure 4.** Proteasome inhibition suppresses ETO-induced genome instability and improves cell survival. (**A**) Example of mitotic spreads from WT primary splenic B-cells treated for 2 hr in G1 with 50 µM ETO ± 1 hr 5 µM BTZ pre-treatment, followed by 24 hr recovery in drug free medium prior to harvesting metaphases. Colored arrows indicate the type of aberrations as quantified and described in **B**. (**B**) Analysis of chromosomal aberrations (Chromosome Breaks (yellow), Chromatid Breaks (orange), Fusions (blue), Radials (purple) and Fragments (green)). 50 metaphases were counted for each condition. (**C**) Example of mitotic spreads from WT primary splenic B-cells treated for 2 hr in G1 with 50 µM ETO ± 1 hr 10 µM MG132 pre-treatment, and then harvested 24 hr after washout for Spectral Karyotyping Analysis (SKY) (Left and right top panels). SKY reveals more complex fusions involving multiple chromosomes, compared to the fusions observable by DAPI staining only (Left and right bottom panels). (**D**) Analysis of chromosomal fusion determined by SKY analysis. Chromosome fusions were counted in 35 mitotic

*Figure 4 continued on next page*

*Figure 4 continued*

spreads per condition and were broken down by how many chromosomes were involved (1, two or greater than 3 (+)) per fusion event per cell. (E) Quantification of percent cells in S phase determined by FACS. WT primary splenic B-cells were treated in G1 for 2 hr with 50 µM ETO ± 1 hr 10 µM MG132 pre-treatment. 16 or 24 hr after washout, cells were pulsed for 30 min with EdU pulse an analyzed by FACS (16 hr NT vs MG132 p=0.012; 16 hr NT v ETO p=0.0485; 24 hr ETO vs pre-MG132 + ETO p=0.049; 24 hr ETO vs. post-MG132 p=0.46; statistical significance calculated by student T-test). (F) Viability of WT primary splenic B-cells as determined using the CellTiter-Glo luminescence assay 48 hr following 2 hr of 50 µM ETO treatment ±pre treatment, co-treatment, or post-treatment with 5 µM BTZ (BTZ v ETO p=0.0006, ETO v pre-BTZ + ETO p=0.015, statistical significance calculated by student T-test). During treatment cells were in the G1 phase of the cell cycle.

proteasome inhibition on ETO-induced genome instability is not cell cycle dependent, and neither

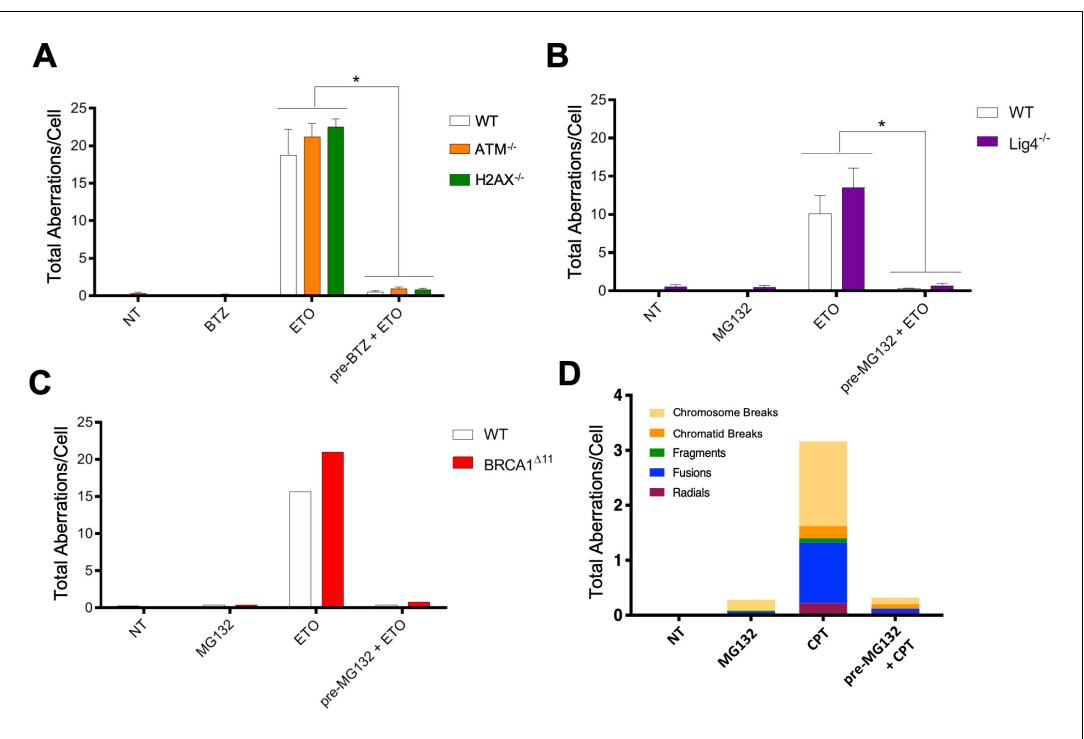

**Figure 5.** Multiple DNA repair pathways contribute to ETO-induced genome instability. (A) Mitotic spread analysis of WT (White), ATM$^{-/-}$ (Orange), and H2AX$^{-/-}$ (Green) primary splenic B-cells treated in G1 for 2 hr with 50 µM ETO ± 1 hr 5 µM BTZ pre-treatment. B-cells were fixed for mitotic spread analysis following a 24 hr recovery at 37°C in drug free media. Total chromosomal aberrations were counted in 50 metaphases (N = 3) and averaged for each genotype. (ETO vs. pre-BTZ + ETO: WT, p=0.006; ATM$^{-/-}$, p=0.0004; H2AX$^{-/-}$, p<0.0001; statistical significance calculated by student T-test). (B) Mitotic spread analysis of WT (White) and Lig4$^{-/-}$ (Purple) primary splenic B-cells treated in G1 for 2 hr with 50 µM ETO ±1 hr 10 µM MG132 pre-treatment. B-cells were fixed for mitotic spread analysis following a 24 hr recovery at 37°C in drug free media. Total chromosomal aberrations were counted in 50 metaphases (N = 3) and averaged for each genotype. (ETO vs pre-MG132 + ETO: WT, p=0.0154; Lig4$^{-/-}$, p=0.0075; statistical significance calculated by student T-test). (C) Mitotic spread analysis of cycling (24 hr post-activation) WT (White) and BRCA1$^{Δ11}$ (Red) primary splenic B-cells treated for 2 hr with 50 µM ETO ± 1 hr 10 µM MG132 pre-treatment. B-cells were fixed for mitotic spread analysis following a 24 hr recovery at 37°C in drug free media. Total chromosomal aberrations were counted in 50 metaphases per condition. (D) Mitotic spread analysis of cycling primary splenic B-cells treated for 2 hr with 12.5 µM CPT ± 1 hr 10 µM MG132 pre-treatment. B-cells were fixed for mitotic spread analysis following a 24 hr recovery at 37°C in drug free media. Total chromosomal aberrations were counted in 50 metaphases per condition.

The online version of this article includes the following figure supplement(s) for figure 5:

**Figure supplement 1.** Multiple DNA repair mechanisms contribute to error-prone repair of ETO-induced damage.

classical NHEJ nor HR appears to be strictly required to generate ETO-induced chromosomal aberrations.

Given the lack of clear dependency on canonical DSB repair pathways, we next explored whether Polθ-mediated alternative end joining could be a major contributor of ETO-induced genome instability. Similar to Lig4- and BRCA1/2-deficiency, however, loss of Polθ (POLQ⁻/⁻) did not significantly attenuate the formation of complex chromosomal rearrangements in ETO-treated cells (*Figure 5—figure supplement 1*). Therefore, multiple DNA repair mechanisms likely contribute to the error-prone repair of ETO-induced DSBs.

## Proteasome inhibitor pre-treatment prevents degradation of TOP1 cleavage complexes and camptothecin-induced genome instability

In addition to preventing the degradation of TOP2ccs, previous studies have shown that proteasome inhibitors are able to suppress ATM-γ-H2AX signaling in response to damage induced by TOP1 poisons like Camptothecin (CPT) (*Lin et al., 2008*), which stabilizes TOP1ccs in the same mechanistic way ETO stabilizes TOP2ccs (*Pommier et al., 2016*). We, therefore, assessed if cells pre-treated with proteasome inhibitors were protected from chromosomal aberrations caused by CPT. Our results showed that MG132 pre-treatment largely suppressed CPT-induced chromosomal aberrations (*Figure 5D*), similar to its effects in ETO-treated cells (*Figure 5A–C*). Thus, inhibiting the proteolytic degradation of either TOP1ccs or TOP2ccs prevents subsequent DNA mis-repair that leads to genome instability.

## The SUMO-targeted ubiquitin ligase RNF4 promotes proteasome-mediated processing of TOP2ccs

Based on the results presented above, it is clear that the degradation of ETO-stabilized TOP2ccs leads to genome instability, while chemical suppression of proteasomal TOP2cc degradation prevents the accumulation and mis-repair of genotoxic DSBs. Recent work has described a SUMO-ubiquitin (Ub) pathway that recognizes DNA-bound topoisomerase-DNA complexes, wherein the SUMO-targeted E3 ubiquitin ligase (STUbL) RNF4 ubiquitinates TOP2 leading to its degradation (*Sun et al., 2019*). We therefore assessed whether interfering with the cell's ability to sense and recruit the proteasome to trapped TOP2ccs would confer a chemo-protective effect similar to cells treated with BTZ and ETO (*Figure 1B*, *Figure 4A–F*). To this end, we treated WT and RNF4⁻/⁻ MEFs (*Hu et al., 2010*) with 10 μM ETO for 1 hr with or without BTZ pre-incubation (1 μM, 1 hr), and probed for γ-H2AX. ETO produced a weakened γ-H2AX response in RNF4⁻/⁻ MEFs compared to WT MEFs (*Figure 6A*), suggesting that RNF4 facilitates proteasomal degradation of TOP2ccs, which in turn generates γ-H2AX. Notably, while BTZ pre-treatment suppressed the ETO-induced γ-H2AX response in WT MEFs (*Figure 6A*), it did not further reduce the γ-H2AX response in RNF4⁻/⁻ MEFs (*Figure 6A*), suggesting that RNF4 is epistatic with the proteasome with regard to TOP2cc degradation. To explore this further, we used END-seq to quantify the number and intensity of TOP2-mediated DSBs in ETO treated RNF4⁻/⁻ and WT primary B-cells. Our results showed that there was a 6-fold reduction of protein-free DSBs detected by END-seq in RNF4⁻/⁻ B-cells compared to WT (*Figure 6B*, top right), while the initial levels of total TOP2-mediated DNA cleavage were similar between the genotypes (*Figure 6B*, top left). This effect was comparable to what we observed by chemically inhibiting the proteasome with BTZ, which reduced the number of protein-free DSBs by ~5 fold (*Figure 6B*, bottom right). These results indicate that RNF4 ubiquitination is critical for proteasome-mediated processing of TOP2ccs into genotoxic protein-free DSBs.

To assess if the loss of RNF4 conferred resistance to ETO, we determined how RNF4 deletion affected short-term cellular viability and long-term colony formation potential in ETO-treated B-cells and MEFs, respectively. We found that RNF4⁻/⁻ cells were significantly more viable and retained higher colony formation capacity after a 1 hr pulse of ETO compared to WT counterparts (*Figure 6C and D*). Consistent with increased viability, RNF4⁻/⁻ B-cells accumulated 60% less chromosomal aberrations after ETO treatment compared to WT B-cells (*Figure 6E*). Taken together, these results indicated that impairing the proteasome response to trapped TOP2ccs reduces the cytotoxicity and genotoxicity of ETO. Notably, pre-treating RNF4⁻/⁻ B-cells with BTZ further reduced aberrant chromosomal repair (*Figure 6E*), suggesting that RNF4-mediated ubiquitination is likely not the only mechanism by which the proteasome is recruited to TOP2ccs.

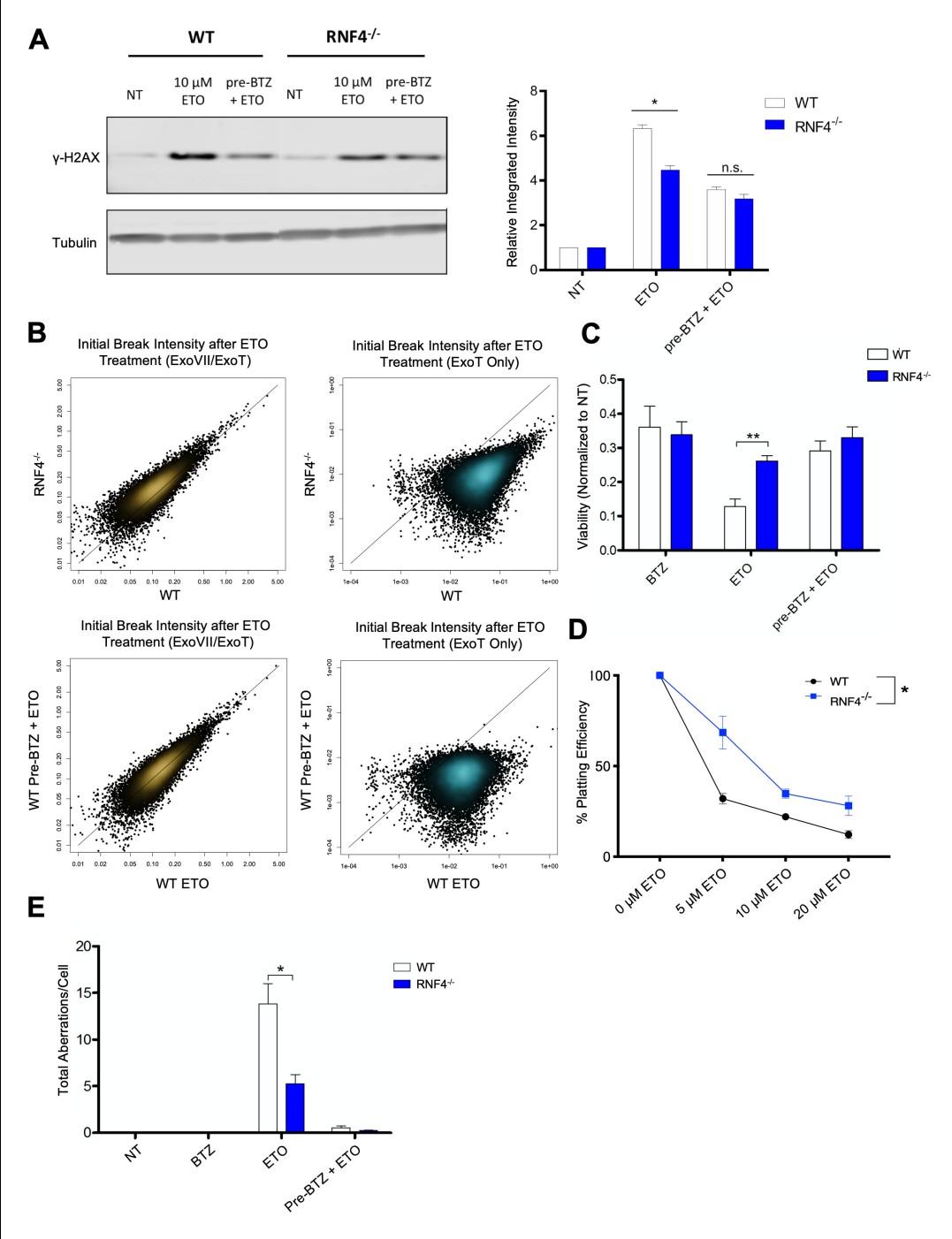

**Figure 6.** The SUMO-targeted Ubiquitin-Ligase RNF4 functions in proteasome mediated processing of TOP2cc.
(A) Example γ-H2AX western blot in WT and RNF4⁻/⁻ MEFs treated with 10 µM ETO for 1 hr ±1 hr 1 µM BTZ pre-treatment (Left Panel). Bar graph quantifying the relative band intensity from γ-H2AX western blots (N = 3) of WT (White) and RNF4⁻/⁻ (Blue) MEFs treated with 10 µM ETO (Right Panel). (WT ETO vs RNF4⁻/⁻ ETO p=0.0018; statistical significance calculated by student T-test). (B) Scatterplots depicting initial intensity of all TOP2-mediated DSBs (ExoVII/ExoT, Top Left) and initial intensity of protein-free DSBs genome-wide (ExoT, Top Right) induced by 2 hr treatment of WT and RNF4⁻/⁻ B cells with 50 µM ETO. Scatterplots of all TOP2-mediated DSBs (ExoVII/ExoT, Bottom Left) and protein-free DSBs genome-wide (ExoT, Bottom Right) in WT primary splenic B-cells treated with 50 µM ETO only vs. 50 µM ETO + 5 µM BTZ pre-treatment. (C) Viability of WT (White) and RNF4⁻/⁻ (Blue) primary splenic B-cells, as determined by the CellTiter-Glo luminescence assay 48 hr following 2 hr 50 µM ETO treatment ±1 hr 5 µM BTZ pre-treatment. (D) Colony formation assay in WT (White) and RNF4⁻/⁻ (Blue) MEFs. Cells
*Figure 6 continued on next page*

*Figure 6 continued*

were seeded in 6 cm plates at 100, 1000 and 10000 cells/plate and pulsed with 5–20 µM ETO for 1 hr and then left to recover and form colonies for 7 days. The number of colonies from duplicate plates were averaged and plotted at each concentration (N = 3) (WT vs RNF4 p=0.0034; statistical significance calculated by ANOVA). (E) Mitotic spread analysis of WT (White) and RNF4[-/-] (Blue) primary splenic B-cells treated in G1 for 2 hr with 50 µM ETO ± 1 hr 5 µM BTZ pre-treatment. B-cells were fixed for mitotic spread analysis following a 24 hr recovery at 37°C in drug free media. Total chromosomal aberrations were counted in 50 metaphases (N = 3) and averaged for each genotype. (WT vs. RNF4[-/-] ETO: p=0.048, statistical significance calculated by student T-test).

## Discussion

In this study, we describe a mechanism by which ETO-induced genotoxicity can be abolished by inhibiting proteasome mediated degradation of TOP2 cleavage complexes (TOP2ccs). In the absence of proteolytic degradation, TOP2 remains enzymatically competent and is able to reseal the protein-linked DSB without invoking a DNA damage response (DDR). However, once degradation commences, a TOP2cc is no longer reversible, and the previously hidden DSB become unmasked, triggering a potent DDR signaling response. Following degradation of a TOP2cc, the resultant protein free DSBs appear to engage multiple repair pathways, including error-free NHEJ (*Gómez-Herreros et al., 2013*; *Gómez-Herreros et al., 2017*; *Figure 7A*). However, due to the large number of lesions induced by ETO, many DSBs remain unrepaired or mis-repaired which can destabilize the genome. In contrast, by preventing proteasome-mediated unmasking of TOP2-mediated DSBs,

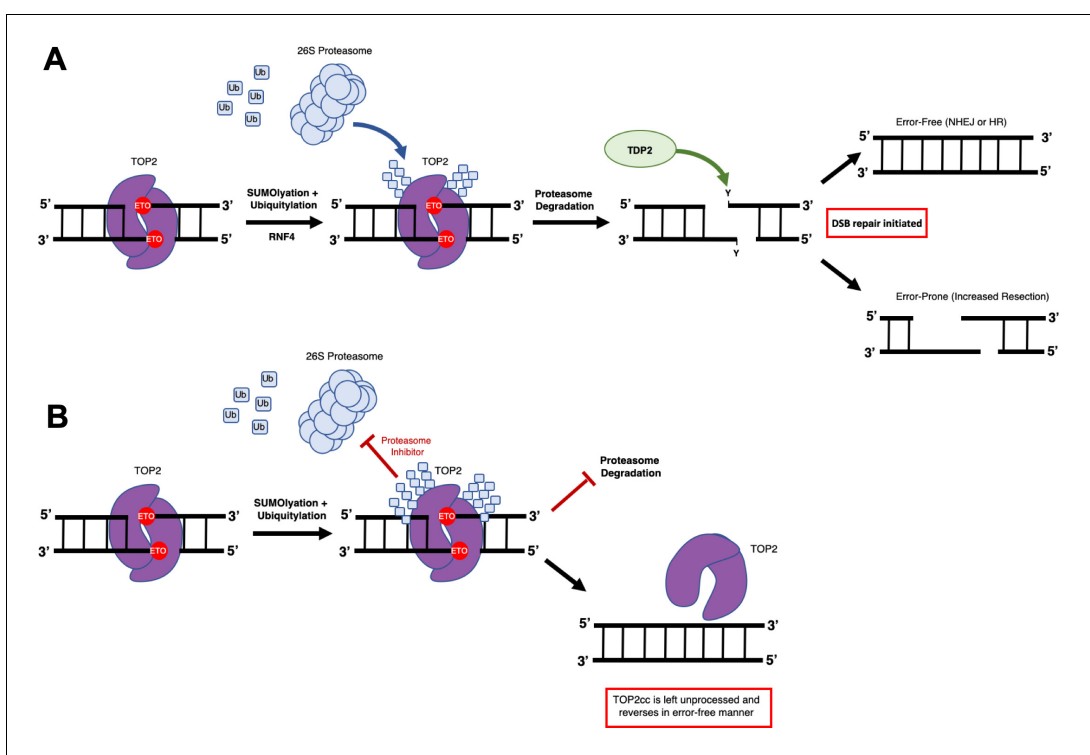

**Figure 7.** Model of proteasome mediated repair of TOP2ccs. (**A**) ETO-stabilized TOP2ccs are targeted for proteasomal degradation by the 26S proteasome via both SUMOylation and Ubiquitination, mediated by the SUMO-targeting ubiquitin ligase RNF4 and possibly other enzymes. Upon degradation of the TOP2 protein within the TOP2cc, the remaining 5'-phosphotyrosyl bonds must be processed by TDP2, generating clean protein-free DSBs with overhangs that can be repaired in a manner that is not always error-free. (**B**) If the proteasome is not recruited due to loss of RNF4-mediated TOP2cc polyubiquitylation or its activity is inhibited chemically, the TOP2cc is not recognized by the DNA repair machinery and the TOP2 protein retains full enzymatic ability to reverse itself once ETO is washed out.

error-prone repair is not engaged, thereby facilitating the reversal of TOP2ccs through completion of the enzymes' catalytic cycle upon drug removal, and preserving genome integrity (*Figure 7B*).

Both proteasome inhibitors (e.g. BTZ) and topoisomerase poisons (e.g. ETO, doxorubicin) are used clinically as therapeutic anti-cancer agents, sometimes in combination (*Cowell and Austin, 2012*; *Dittus et al., 2018*; *Manasanch and Orlowski, 2017*; *Nitiss, 2009b*; *Palumbo et al., 2008*; *Thomas et al., 2017*). Notably, prior studies have shown that proteasome inhibitors potentiate the cytotoxicity of TOP2 poisons (*Ceruti et al., 2006*; *Destanovic et al., 2018*; *Dittus et al., 2018*; *Lee et al., 2016*; *von Metzler et al., 2009*), which at first glance appeared to be at odds with our observation that proteasome inhibition enhances genome stability and survival of ETO-treated cells. We addressed this apparent discrepancy by showing that the timing of proteasome inhibition has a remarkable impact on the outcome of ETO treatment. Inhibiting proteasomal function prior to or during ETO administration significantly improved genome stability by minimizing DSB mis-repair associated with TOP2 proteolysis. However, inhibiting the proteasome after ETO treatment did not effectively protect cells from ETO-induced genome instability, presumably because proteasome-mediated degradation of TOP2ccs is an extremely fast process. Taken together, our study suggests that caution needs to be taken when scheduling therapeutic regimens combining topoisomerase poisons with proteasome inhibitors.

Our analysis also revealed that TOP2-mediated DSBs, especially ones that are not promptly repaired, undergo extensive DNA end-resection. We interpreted this as an indicator of active DNA repair occurring at TOP2ccs that have been partially or completely degraded by the proteasome. By contrast, when proteasomal degradation was inhibited by BTZ, we could only detect trace levels of DNA end-resection. This suggests the possibility that MRE11, which is required for DNA end-resection, may not recognize unprocessed TOP2. Similar to TOP2, the meiotic recombination initiator SPO11 forms protein-linked DSBs, which are thought to trigger MRE11 nuclease removal of SPO11 (*Neale et al., 2005*). However, recent studies reveal that mammalian meiotic cells accumulate significant levels of DNA-bound SPO11, which could, in principal be partially proteolyzed (*Paiano et al., 2020*). While the mechanism by which TOP2 (and perhaps SPO11) shields its associated DSB from early detection by DDR surveillance factors is unclear, our results highlight that γ-H2AX is only a reliable marker for proteolytically degraded TOP2ccs. In contrast, END-seq can be used to distinguish intact TOP2ccs, partially proteolyzed TOP2ccs, as well as fully processed protein-free DSBs.

Finally, we found that the SUMO-targeted ubiquitin ligase RNF4 played a critical role in recruiting the proteasome to degrade TOP2ccs, confirming a report by another group (*Sun et al., 2019*). The loss of RNF4 afforded cells significant protection from the genotoxic effects of ETO, further highlighting how impairing proteasome-mediated degradation of TOP2ccs can preserve genome integrity. However, RNF4 deletion did not prevent ETO-induced chromosomal aberrations or blunt the γ-H2AX response as efficiently as chemical proteasome inhibition. This indicates the likely existence of additional ubiquitin ligases that are functionally redundant to RNF4. Nevertheless, our data suggests that RNF4 could be potentially used as a biomarker to predict the effectiveness of chemotherapeutic regimens that incorporate topoisomerase poisons. In addition, de-regulation of RNF4 may represent a potential mechanism for tumors to acquire resistance to topoisomerase poisons by favoring TOP2cc reversal over degradation.

## Materials and methods

### Key resources table

| Reagent type (Species or Source) | Designation | Source | Identifier | Add. info |
|---|---|---|---|---|
| Antibody | Anti-phospho-Histone H2A.X (Ser139) (mouse monoclonal) | Millipore | Cat# JBW301; RRID:AB_309864 | (1:5000 - 1:10000) |
| Antibody | Anti-Tubulin (mouse monoclonal) | Sigma | Cat# T-5168 RRID:AB_477585 | (1:5000) |
| Antibody | Anti-p53 (1CA2) (mouse monoclonal) | Cell Signaling | Cat# 2524 RRID:AB_331743 | (1:2000) |

*Continued on next page*

*Continued*

| Reagent type (Species or Source) | Designation | Source | Identifier | Add. info |
|---|---|---|---|---|
| Antibody | Anti-Vinculin (rabbit polyclonal) | Cell Signaling | Cat# 4650 RRID:AB_10559207 | (1:2000) |
| Antibody | Anti-Top2b (rabbit polyclonal) | Novus | Cat# NB100-10842 RRID:AB_1522570 | (1:5000) |
| Antibody | Anti-Rabbit Li-Cor (goat polyclonal) | Li-Cor | Cat# 926–32211; RRID:AB_621843 | (1:10000) |
| Antibody | Anti-mouse IgG1 Alexa 488 (goat polyclonal) | Invitrogen | Cat# A-21121; RRID:AB_141514 | (1:10000) |
| Antibody | Anti-H2AX (rabbit polycolonal) | Abcam | Cat# ab11175 RRID:AB_297814 | (1:5000) |
| Chemical compound, Drug | Etoposide | Sigma | Cat# E1383 | 10–50 µM |
| Chemical compound, Drug | Bortezomib | Millipore | Cat# 179324-69-7 | 1–5 µM |
| Chemical compound, Drug | MG132 | Sigma | Cat# 1211877-36-9 | 10 µM |
| Chemical compound, Drug | Camptothecin | Sigma | Cat# 7689-03-4 | 12.5 µM |
| Chemical compound, Drug | Ixaomib | Selleckchem | Cat# S2180 | 10 µM |
| Chemical compound, Drug | Eponomicin | Sigma | Cat# 134381-21-8 | 2 µM |
| Chemical compound, Drug | Doxycycline hyclate | Sigma | Cat# D9891 | 1 µg/ml |
| Chemical compound, Drug | Imatinib mesylate (STI571) | Selleckchem | Cat# S1026 | |
| Cell Line (*M. musculus*) | pre-B cell lines | *Bredemeyer et al., 2006* | | |
| Cell Line (*M. musculus*) | RNF4$^{-/-}$ MEFs | Generated by Gary Lyons, *Hu et al., 2010* | | |
| Biological Sample (*M. musculus*) | Primary splenic B-cells | | Freshly isolated from *M. musculus* spleen | |
| Strain (*M. musculus*) | C57BL/6NCr mice (WT) | Charles River | Strain code# 027 | |
| Strain (*M. musculus*) | Conditional BRCA1-Δ11f/f CD19-cre expressing mice | NCI mouse repository | | |
| Strain (*M. musculus*) | H2AX$^{-/-}$ mice | *Celeste et al., 2002* | | |
| Strain (*M. musculus*) | Conditional Lig4f/f expressing ERT2-Cre mice | Provided by P. Mckinnon | | |
| Strain (*M. musculus*) | ATM$^{-/-}$ mice | Provided by A. Wynshaw-Boris | | |
| Strain (*M. musculus*) | Conditional RNF4 f/f mice | Provided by S. Bunting | | |
| Strain (*M. musculus*) | Conditional BRCA2 f/f CD19-cre expressing mice | Provided by S. Sharan | | |
| Strain (*M. musculus*) | PolQ$^{-/-}$ mice | Provided by A. D'Andrea | | |
| Recombinant DNA reagent | YFP$^u$ Degron reporter (plasmid) | Provided by A. Weissman *Bence et al., 2005*; *Bence et al., 2001* | | (0.5 ng/µL) |

*Continued on next page*

*Continued*

| Reagent type (Species or Source) | Designation | Source | Identifier | Add. info |
|---|---|---|---|---|
| Peptide, recombinant protein | Lipopolysaccharide (LPS) | Sigma | Cat# L-2630 | 25 µg/ml |
| Peptide, recombinant protein | IL-4 from mouse, Interleukin-4, recombinant | Sigma | Cat# I-1020 | 5 ng/ml |
| Peptide, recombinant protein | RP105 | BD Biosciences | Cat # 552128 | 0.5 µg/ml |
| Peptide, recombinant protein | Cy3-labeled (CCCTAA) peptide nucleic acid probe | PNA Bio | Cat# F1002 | |
| Peptide, recombinant protein | Puregene Proteinase K solution | Qiagen | Cat# 19133 | 170 µL |
| Peptide, recombinant protein | Puregene RNaseA | Qiagen | Cat# 19101 | 50 µL |
| Peptide, recombinant protein | T4 DNA polymerase | NEB | Cat# M0203L | 15 U |
| Peptide, recombinant protein | Klenow fragment | NEB | Cat# M0210M | 5 U |
| Peptide, recombinant protein | T4 polynucleotide kinase | NEB | Cat# M0201L | 15 U |
| Peptide, recombinant protein | Klenow exo- fragment | NEB | Cat# M0212M | 15 U |
| Peptide, recombinant protein | Quick Ligase | NEB | Cat# M2200L | |
| Peptide, recombinant protein | USER enzyme | NEB | Cat# M5508L | |
| Peptide, recombinant protein | beta-agarase I | NEB | Cat# M0392S | 1.5 µL |
| Commercial Assay Kit | CHEF Mammalian Genomic DNA plug kit | Bio-Rad | Cat# 1703591 | |
| Commercial Assay Kit | Click-IT EdU Alexa Fluor 488 Flow Cytometry Assay Kit | ThermoFisher | Cat# C10425 | |
| Commercial Assay Kit | KAPA Library Quantification Kit | Kapa Biosciences | Cat# KK4824 | |
| Commercial Assay Kit | Celltiter-Glo Cell Viability Assay Kit | Promega | Cat# G7571 | |
| Other | Streptavidin beads MyOne C1 | ThermoFisher | Cat# 650–01 | 35 µL |
| Other | Anti-CD43 (Ly-48) MicroBeads (mouse) | Miltenyi Biotech | Cat# 130-049-80 | |
| Other | Glycogen (20 mg/ml) | Roche | Cat# 10901393001 | 1 µL |
| Other | Kapa HiFi Hot Start Ready mix | Kapa Biosciences | Cat# KK2502 | |

*Continued on next page*

*Continued*

| Reagent type (Species or Source) | Designation | Source | Identifier | Add. info |
|---|---|---|---|---|
| Other | Agencourt AM Pure XP beads | Beckman Coulter | Cat# A63881 | |
| Other | IrysPrep Lysis Buffer | BioNano Genomics | Cat# 20270 | |
| Other | X-tremeGene 9 Transfection Reagent | Sigma | Cat # XTG9-RO | |
| Sequence-based reagent | END-seq hairpin adaptor 1, 5'-Phos-GATCGGAAGAGCGTCGTGT AGGGAAAGAGTGUU[Biotin-dT] U[Biotin-dT]UUACACTCTTTCCCTACACGA CGCTCTTCCGATC*T-3' | *Canela et al., 2016* | END-seq sequence adapter | |
| Sequence-based reagent | END-seq hairpin adaptor 2, 5'-Phos-GATCGGAAGAGCACACGTC UUUUUUUUAGACGTGTGCT CTTCCGATC*T-3' | *Canela et al., 2016* | END-seq sequence adapter | |
| Sequence-based reagent | TruSeq barcoded primer p5, AATGATACGGCGACCACCGAGATCTA CACNNNNNNNNACACTCTTTCCCTA CACGACGCTCTTCCGATC*T | *Canela et al., 2019* | END-seq sequence primer | |
| Sequence-based reagent | TruSeq barcoded primer p7, CAAGCAGAAGACGGCATACGAGA NNNNNNNGTGACTGGAGTTCAG ACGTGTGCTCTTCCGATC*T | *Canela et al., 2019* | END-seq sequence primer | |
| Software | Prism (7) | GraphPad | https://www.graphpad.com/scientific-software/prism/ RRID:SCR_002798 | |
| Software | Bowtie 1.1.2 | *Langmead et al., 2009* | https://sourceforge.net/projects/bowtie-bio/files/bowtie/1.1.2/ RRID:SCR_005476 | |
| Software | MACS 1.4.3 | *Zhang et al., 2008* | https://pypi.python.org/pypi/MACS/1.4.3 | |
| Software | UCSC database | *Karolchik et al., 2004* | https://genome.ucsc.edu RRID:SCR_005780 | |
| Software | UCSC Genome Browser | *Kent et al., 2002* | https://genome.ucsc.edu RRID:SCR_005780 | |
| Software | Bedtools | *Quinlan and Hall, 2010* | https://github.com/arq5x/bedtools2 RRID:SCR_006646 | |
| Software | Samtools | *Li et al., 2009* | https://github.com/samtools/samtools RRID:SCR_002105 | |
| Software | Knight-Ruiz algorithm | *Rao et al., 2014* | https://academic.oup.com/imajna/article-lookup/DOI: 10.1093/imanum/drs019 | |
| Software | R | *R Development Core Team (2008)* | https://www.r-project.org/ RRID:SCR_001905 | |
| Software | FlowJo (10.1) | FlowJo LLC | https://www.flowjo.com/ RRID:SCR_008520 | |
| Software | Metapher ISIS | Metapher Systems | | |
| Software | hiSKY 7.2.7 | ADS Biotech | | |

## Mice

C57BL/6 WT (NCI mouse repository), POLQ$^{-/-}$ (provided by A. D'Andrea), H2AX$^{-/-}$ (*Celeste et al., 2002*), ATM$^{-/-}$ (provided by A. Wynshaw-Boris) and conditional Lig4$^{-/-}$ (Lig4$^{f/f;ERT2-Cre}$, provided by P. McKinnon), BRCA1$^{\Delta11}$BRCA1$^{f/f;CD19Cre}$, *Zong et al. (2019)*, BRCA2$^{-/-}$ (BRCA2$^{f/f;CD19Cre}$, provided by

S. Sharan) and RNF4$^{-/-}$ (RNF4$^{f/f;CD19Cre}$) mice between 8 and 18 weeks of age were used to prepare single cell suspensions of primary splenic B-cells. All animal experiments were approved by the NCI Animal Care and Use Committee (Protocol Numbers: EIB-064–3 and 17–042).

## Cell culture methods

Mature resting B-cells were isolated from mouse spleen with anti-CD43 MicroBeads (Miltenyi Biotech). B-cells were activated with LPS (25 µg/ml; Sigma), IL-4 (5 ng/ml; Sigma-Aldrich) and RP105 (0.5 µg/ml; BD Biosciences) for 12 hr as described (Barlow et al., 2013; Callén et al., 2007). Lymphocyte separation media (Corning) was used after harvesting the B-cells to separate dead cells from live cells prior to use for END-seq protocol. Cellular Viability was measured using CellTiter-Glo Cell Viability Assay Kit (Promega) according to manufactures directions. Luminescence was quantified using a Tecan Infinite M200 Pro plate reader. B-cells were pre-treated with proteasome inhibitor, either 10 µM MG132 (Sigma-Aldrich) or 5 µM Bortezomib (Millipore) for 1 hr then co-treated for an additional 2 hr with 50 µM Etoposide (E1383, Sigma-Aldrich). Following the ETO pulse, the cells were either harvested in the presence of the drugs, or washed three times in ice-cold drug free media (3 × 5 min spin at 1500 rpm and 4°C to pellet B-cells between washes; 15–20 min total time to complete washout), and returned to 37°C to recover in fresh drug free media, until they were harvested at various timepoints post-washout for END-seq, western blot, Immunofluorescence, ICE assay, or for mitotic spread analysis as described in Figure 1A. Asynchronous WT B-cells were treated 24 hr post-activation for 2 hr with 50 µM ETO or 12.5 µM Camptothecin (Sigma-Aldrich) ±1 hr 10 µM MG132 pre-treatment, then fixed following a 24 hr recovery in drug free media for mitotic spread analysis. Abelson-transformed pre-B cells (Bredemeyer et al., 2006) were retrovirally transduced with the tetracycline-inducible ER-AsiSI, pTRE3G-HA-ER-AsiSI as previously described (Callen et al., 2013). WT and RNF4$^{-/-}$ MEFs (generated by Gary Lyons) (Hu et al., 2010) were cultured in DMEM (Invitrogen) + 10% FBS and 1% PenStrep (Invitrogen). MEFs were treated with 5–20 µM ETO for 1 hr ±1 µM BTZ 1 hr pretreatment. For END-seq experiments that used Zinc Finger Nuclease (ZFN) spike-in normalization, induction of ZFN was done in G1-arrested Lig4$^{-/-}$ pre B cells (Bredemeyer et al., 2006) by treating with imatinib for 48 hr and 1 µg/ml Doxycycline during the last 24 hr as described (Canela et al., 2016; Bredemeyer et al., 2006).

WT and Lig4-/- mouse pre B cell lines were provided by B. Sleckman. WT and RNF4$^{-/-}$ mouse embryonic fibroblasts (MEFs) were provided by S. Bunting. These cell lines are negative for mycoplasma.

## γ-H2AX and pan-p53 western blot

Western blotting for γ-H2AX and pan-p53 was performed using γ-H2AX antibody (Millipore) at 1:5000 dilution and p53 antibody (Cell Signaling) at 1:2000 dilution. Antibodies recognizing tubulin (Sigma-Aldrich, 1:5000), H2AX (Millipore, 1:5000) and vinculin (Cell Signaling, 1:2000) were used to control for protein loading. Image analysis was done using Li-Cor Odyssey CLx to quantify band intensity.

## In vitro of complex (ICE) Assay

Topoisomerase II DNA-protein complexes (TOP2-DPCs) were isolated and detected using in vivo complex of enzyme (ICE) assay as previously described (Anand et al., 2018). Briefly, 5 million cells were lysed in sarkosyl solution (1% w/v) after treatment. Cell lysates were sheared through a 25 g 5/8 needle (10 strokes) to reduce the viscosity of DNA and layered onto CsCl solution (150% w/v), followed by centrifugation in NVT 65.2 rotor (Beckman coulter) at 42,000 RPM for 20 hr at 25°C. The resulting pellet containing nucleic acids and TOP2-DPCs was obtained and dissolved in TE buffer. The samples were subjected to immunoblotting (slot blot) with anti-mouse TOP2β antibody (Novus). 2 µg of DNA is applied per sample. TOP2-DPCs were quantified by densitometric analysis using ImageJ.

## Metaphase spread, FISH and SKY analysis

5 million cells were harvested 24 hr after drugs were washed out for metaphase analysis as described (Callén et al., 2007). Quantitative FISH analysis using a Cy3-labeled (CCCTAA) peptide nucleic acid probe (Applied Biosystems) was performed as described previously (Callén et al., 2007). Telomere

length measurements were performed on least 15 metaphases for each cell type. DAPI chromosome and Cy3 telomere images were acquired with a constant exposure time that ensured all captured fluorescent signals were within the linear range using Metafer Software on a Zeiss Axios Imager Z2 Microscope. Image analysis was done in the Metafer ISIS software. SKY was performed and analyzed as previously described, using the hiSKY 7.2.7 Software (ASI) on a Leica DMRXA Microscope (*Liyanage et al., 1996*). A minimum of 35 metaphases were imaged and analyzed.

## YFP$^u$degron quantification

WT eHAP cells were transfected for 48 hr using Xtreme Gene 9 transfection reagent with YFP$^u$ Degron reporter (0.5 ng/μL) as described (*Bence et al., 2005*; *Bence et al., 2001*). eHAP cells were then treated 48 hr post-transfection with 1 μM BTZ and 5 μM MG132 for 2 hr to mimic the total amount of time the proteasome inhibitors are incubated in primary B-cells, after which the cells were washed three times and left to recover at 37°C for up to 18 hr. Cells were fixed at 2 hr, 6 hr, and 18 hr after washout with 4% PFA and YFP expression was measured on a FACS CantoII (BD biosciences).

## EdU staining

To measure DNA synthesis, primary B-cell cultures were stimulated for 12 hr, 24 hr, 28 hr or 36 hr pulsed with 10 μM of EdU (5-ethynyl-2′-deoxyuridine) for 30 min at 37°C and stained using the Click-IT EdU Alexa Fluor 488 Flow Cytometry Assay Kit according to the manufacturer's specifications (Thermo Fisher C10425). Samples were acquired on a FACS CantoII (BD biosciences).

## Immunofluorescence

Immunofluorescence staining with γ-H2AX antibody (Millipore, 1:10,000) was performed in parallel in WT pre B cells (*Bredemeyer et al., 2006*) to verify DSB induction.

## END-seq

Single cell suspensions of primary B-cells (15–20 million) were untreated or treated with drugs to inhibit the proteasome and/or etoposide as indicated in the cell culture methods section. Primary B-cells were washed twice in cold PBS and resuspended in cold cell suspension buffer (Bio-Rad CHEF Mammalian Genomic DNA plug kit), equilibrated for 5 min at room temperature, mixed with 2% melted CleanCut agarose (Bio-Rad CHEF Mammalian Genomic DNA plug kit) prewarmed at 37°C for a final concentration of 0.75%, and transferred immediately into plug molds and let them solidify at 4°C for 20 min. Embedded cells were lysed and digested using Proteinase K (50°C, 1 hr then 37°C for 7 hr). Etoposide and drugs were maintained at the same experimental concentrations in all the steps until digestion with Proteinase K. Plugs were rinsed in TE buffer and treated with RNaseA at 37°C, 1 hr. The next enzymatic reactions were performed with the DNA in agarose plugs to prevent shearing. DNA ends were blunted for 1 hr at 37°C with Exonuclease VII (ExoVII) followed by Exonuclease T (ExoT) (NEB) for 1 hr at 25°C to detect all TOP2-mediated DSBs (protein-linked and protein-free), or only with Exonuclease T (NEB) for 1 hr at 25°C to detect only protein-free DSBs. After blunting, A-tailing was performed to attach dA to the free 3′-OH, followed by ligation of ''END-seq hairpin adaptor 1,'' listed in reagents section, using NEB Quick Ligase. Agarose plugs were then melted and dialyzed, and DNA was sonicated to a median shear length of 170 bp using Covaris S220 sonicator for 4 min at 10% duty cycle, peak incident power 175, 200 cycles per burst, at 4°C. DNA was ethanol-precipitated and dissolved in 70 ml TE buffer. Biotinylated DNA was isolated using MyOne Streptavidin C1 Beads (Thermo Fisher #650–01), followed by end repair (dNTPs, T4 polymerase (NEB), Klenow (NEB), T4 PNK) and dA-tailing (Klenow exo- (NEB), dATP). The second end was ligated to ''END-seq hairpin adaptor 2'' using NEB Quick Ligase. Hairpins were digested using USER (NEB), and the resulting DNA fragments were PCR amplified using TruSeq barcoded primer p5, AATGATACGGCGACCACCGAGATCTACACNNNNNNNNNACACTCTTTCCCTACAC-GACGCTCTTCCGATC*T and TruSeq barcoded primer p7, CAAGCAGAAGACGGCATACGAGA NNNNNNNGTGACTGGAGTTCAGACGTGTGCTCTTCCGATC*T, (NNNNNNNN represents barcode a * a phosphothiorate bond) listed in reagents. PCR fragments were isolated by size selection from agarose gel, selecting 200–500 bp fragments followed by DNA purification using QIAquick Gel Extraction Kit. Libraries were quantified using KAPA Library Quantification Kit and sequenced using

Illumina NextSeq 500 or 550. A detailed END-seq protocol can be found in *Canela et al. (2017)*, as well as more information about the ExoVII and ExoT processing of TOP2-mediated DSBs in *Canela et al. (2019)*.

## Quantification and statistical analysis

### Genome alignment
END-seq single end reads were aligned to the mouse genome (GRCm38p2/mm10) using Bowtie v1.1.2 (*Langmead and Salzberg, 2012*) with parameters (-n 3 k 1 l 50) and alignment files were generated and sorted using SAMtools (*Li et al., 2009*) and BEDtools (*Quinlan and Hall, 2010*).

### Peak calling
Peaks were called using MACS 1.4.3 (*Zhang et al., 2008*) with the parameters: –nolambda,–nomodel and–keep-dup = all (keep all redundant reads). For ETO-treated END-seq data peak calling, we used the corresponding non-treated samples as control, keeping >10 fold-enriched peaks. The peaks within blacklisted regions (https://sites.google.com/site/anshulkundaje/projects/blacklists) were filtered. For primary B-cell data, in order to perform a direct comparison of the breaks across experiments, peak calling was done on the END-seq (ExoT+ExoVII) data presented in *Figure 2* (carried out after 2 hr of ETO) and in *Figure 2—figure supplement 1*. As the END-seq signal is very reproducible between replicates (*Figure 2—figure supplement 1*), all of the analyses thereafter including persistence, reversibility and resection presented in *Figures 2* and *3*, utilized the peak dataset from replicate 1. Analysis using the peak dataset from replicate two is presented in *Figure 2—figure supplement 1* and *Figure 3—figure supplement 1*.

### END-seq data analysis
For comparisons between different genotypes and treatments, END-seq signal was calculated, as RPKM, within END-seq peaks, for the corresponding cell type. For END-seq data, all reads were counted. For END-seq data, where cells with ZFN break were spiked-in to the library at a 1:40 (2.5%) ratio, the RPKM value for each peak was divided by the signal around the spiked-in breaks and then multiplied by 2.5, to get values as cell-percentage.

### Resection quantification
To quantify the width of maximum resection endpoint, a sliding window containing 10 50 bp bins was used to start from the summit of each break, out to 2.5 kb on either side of the summit to detect the END-seq signal around the break summit. Maximum END-seq signal for 50 bp bins within 2.5 kb ~5 kb around break summit was used as background. When more than half of the bins within this sliding window had an RPKM values equal to or lower than the background, then the last bin within the window which had a detectable signal over background is regarded as the maximum resection endpoint from the break summit. Only breaks having >100 bp resection signal were considered for the comparison of resection length in *Figure 3*.

### Classification of TOP2-mediated DSBs
The fraction of the different TOP2-mediated DSB species (Reversible TOP2cc, Irreversible TOP2cc and protein-free DSB) were calculated as follows: First, ExoT END-seq signal was divided by the END-seq signal (using ExoVII+ExoT) for all TOP2-mediated DSB sites (determined by peak calling as described above), to obtain the protein-free DSB fraction ([DSB]) (*Chase and Richardson, 1974*). Similarly, TOP2cc fraction, [TOP2CC], was defined as [TOP2CC]=1-[DSB] fraction for each lesion. To calculate the reversible TOP2cc fraction ([TOP2CC]$_R$), we subtracted the data obtained from the END-seq ETO washout experiment (see above), from the data obtained from END-seq ETO experiment without washout, to get [TOP2CC]$_R$. Finally, the irreversible TOP2cc fraction ([TOP2CC]$_I$) was defined as [TOP2CC]$_I$ = 1-[DSB]-[TOP2CC]$_R$. A more detailed description of this analysis can be found in *Canela et al. (2019)*.

### Data visualization
Aligned-reads bed files were first converted to bedgraph files using bedtools genomecov (*Quinlan and Hall, 2010*) following by bedGraphToBigWig to make a bigwig file (*Kent et al., 2002*).

Visualization of genomic profiles was done by the UCSC browser (*Kent et al., 2002*). Genome browser profiles were normalized to the library size (RPM) and spike-in.

## Statistical analysis

Statistical analysis was performed using R version 3.5.0 (http://www.r-project.org). The statistical tests are reported in the figure legend and main text.

## Acknowledgements

We thank Keith Caldecott for comments on the manuscript, Allan Weissman for providing the YFP$^u$ Degron, Rachel Burga for support with statistical analysis and generating the graphic schematic and Jennifer Wise and Kelly Smith for assistance with animal work. The AN laboratory is supported by the Intramural Research Program of the NIH, a Department of Defense awards W81XWH-16-1-599) and W81XWH-19-1-0712 and an NIH Intramural FLEX Award.

## Additional information

### Funding

| Funder | Grant reference number | Author |
| --- | --- | --- |
| National Institutes of Health | Intramural Research Program | André Nussenzweig |
| Ellison Medical Foundation | Senior Scholar in Aging Award AG-SS- 2633-11 | André Nussenzweig |
| U.S. Department of Defense | Idea Expansion Award W81XWH-15-2-006 | André Nussenzweig |
| U.S. Department of Defense | Idea Breakthrough Award W81XWH-16-1-599 | André Nussenzweig |
| Alex's Lemonade Stand Foundation | | André Nussenzweig |
| National Institutes of Health | Intramural FLEX Award | André Nussenzweig |

The funders had no role in study design, data collection and interpretation, or the decision to submit the work for publication.

### Author contributions

Nicholas Sciascia, Conceptualization, Data curation, Software, Formal analysis, Validation, Investigation, Visualization, Methodology, Project administration; Wei Wu, Data curation, Software, Formal analysis, Visualization, Methodology; Dali Zong, Formal analysis, Investigation, Methodology, Writing - original draft, Writing - review and editing; Yilun Sun, Formal analysis, Investigation, Visualization; Nancy Wong, Investigation; Sam John, Writing - original draft, Writing - review and editing; Darawalee Wangsa, Data curation, Formal analysis, Investigation, Visualization; Thomas Ried, Formal analysis, Methodology; Samuel F Bunting, Resources, Provided the RNF4 conditional knockout mice; Yves Pommier, Conceptualization, Methodology; André Nussenzweig, Conceptualization, Resources, Supervision, Funding acquisition, Validation, Methodology, Writing - original draft, Project administration, Writing - review and editing

### Author ORCIDs

Nicholas Sciascia (ID) https://orcid.org/0000-0003-4169-4929
André Nussenzweig (ID) https://orcid.org/0000-0002-8952-7268

### Ethics

Animal experimentation: All mouse breeding and experimentation followed protocols approved by the National Institutes of Health Institutional Animal Care and Use Committee (Protocol Numbers: EIB-064-3 and 17-042).

Decision letter and Author response

Decision letter https://doi.org/10.7554/eLife.53447.sa1

Author response https://doi.org/10.7554/eLife.53447.sa2

## Additional files

### Supplementary files

- Supplementary file 1. Table of experimental replicates for each figure panel.

- Transparent reporting form

### Data availability

Sequencing data has been deposited in GEO under the accession code GSE140372.

The following dataset was generated:

| Author(s) | Year | Dataset title | Dataset URL | Database and Identifier |
|---|---|---|---|---|
| Sciascia N, Wu W, Zong D, Sun Y, Wong N, Wangsa D, John S, Ried T, Bunting S, Pommier Y, Nussenzweig A | 2019 | Suppressing Proteasome Mediated Processing of Topoisomerase II DNA-Protein Adducts Preserves Genome Integrity | https://www.ncbi.nlm.nih.gov/geo/query/acc.cgi?acc=GSE140372 | NCBI Gene Expression Omnibus, GSE140372 |

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
