## [Decision Letter]

Thank you for submitting your article "Suppressing proteasome mediated processing of topoisomerase II DNA-Protein adducts preserves genome integrity" for consideration by *eLife*. Your article has been reviewed by three peer reviewers, one of whom is a member of our Board of Reviewing Editors, and the evaluation has been overseen by Maureen Murphy as the Senior Editor. The following individual involved in review of your submission has agreed to reveal their identity: Wolf-Dietrich Heyer (Reviewer #1).

The reviewers have discussed the reviews with one another and the Reviewing Editor has drafted this decision to help you prepare a revised submission.

Title: The title should be modified as follows: "Suppressing proteasome mediated processing of topoisomerase II DNA-Protein complexes preserves genome integrity"

Summary:

In this manuscript the authors follow-up on previous observations regarding the fate of TOP2 cleavage complexes and the role of the proteasome in processing such adducts. The authors confirm previous observations that TOP2 cleavage complexes are readily induced by etoposide eliciting a DNA damage response as measured by accumulation of histone γ-H2AX. They also confirm the previous observation that proteasome inhibitors suppress the DNA damage response in response to etoposide using primary B-cells. Interestingly the authors find this effect still occurs in B cells isolated from mice with ATM^-/-^ and H2AX^-/-^, indicating that the repair may be independent of HR or NHEJ. In cycling MEFs from the RNF4 knockout mouse, deletion of RNF4 is epistatic with proteasome inhibition, supporting the premise that RNF4 plays a role in the degradation of TOP2 complexes.

Significant points:

1) The authors extend upon previous insights and demonstrate that pre- or co-treatment of etoposide with proteasome inhibitors completely suppresses the typical chromosomal rearrangements associated with etoposide treatment. This suppression is the likely consequence of a defect in processing etoposide-induced DNA damage by DSB end resection.

2) The authors demonstrate that the observed chromosomal rearrangements after etoposide treatment are independent of Ligase IV, excluding NHEJ as a mechanism, and BRCA1, suggesting independence of homologous recombination.

3) In an orthogonal genetic approach, the authors show that the SUMO-targeted ubiquitin ligase RNF4 is required for required for proteasome-mediated turnover of TOP2 cleavage complexes and acts epistatically with the proteasome in processing TOP2 cleavage complexes to result in induction of the DNA damage response, loos of viability, and generation of chromosomal aberrations.

4) The conclusions about the role of the proteasome in processing etoposide-induced DNA damage are relevant and potentially impactful, as both etoposide and proteasome inhibitors are used in the clinic alone and in combination. The mechanistic insights gained through this study could provide possible explanation for the variability in efficacy observed for etoposide and has implications for the staging in protocols involving both types of drugs.

Essential revisions:

The main interpretations are, in general, supported by the data, though there are some concerns/exceptions noted below.

1) According to the present draft, many of the experiments appear to have only been done once: for example Figure 1B, C, D, F; Figure 2D and G; Figure 3B; Figure 4B and D; Figure 5B and C; Figure 6A. The authors need to more accurately state how many biological and technical replicates were done, and how exactly statistical analysis was done. Whereas a table stating such was provided to the reviewers, this information needs to be now included in clear manner in a revised manuscript.

2) Figure 1D is a slot blot of ICE assay fractions. It is unclear what the difference is between ETO and ETO + 0 hr washout; or between pre-MG132 + ETO and pre-MG132 + ETO + 0 hr washout. It seems this was done only once. Why? In the reference cited for the method it recommends doing the ICE assay N=3. In order to make quantitative conclusions from this type of data, the experiment must be carried out several times (at least three) so that statistical testing can be applied. And why has only TOP2B been probed for and not TOP2A? If this is because the cells used express very little TOP2A then this should be stated (and preferably demonstrated).

3) In several cases there are not enough details about methods, and it is requested that the authors carefully go through the manuscript to ensure there are enough details that will enable others to reproduce their findings:

a) The section on DNA END-seq is novel, yet it is not clearly detailed enough for others to repeat.

b) In the text it says ETO was washed out "immediately", but in the Materials and methods section, it says cells were washed three times in drug free media. Washing three times would take time. The authors should state how long this took and at what temperature was the washing done. These details are necessary to interpret the reported data.

c) Figure 3A and C – not clear images, and not clear how many replicates. Please clarify how long after the drug treatments were the mitotic spreads produced.

4) In Figure 1, the authors use primary mouse B-cells to show that pre-treatment with proteasome inhibitors suppresses the γ-H2AX response to Etoposide. After washout of the drugs, TOP2 trapped complexes quickly turn over, and the authors suspect that this is due to resumption of TOP2 activity rather than reactivation of the proteasome. To support this conclusion, the authors conduct an experiment in eHAP cells (Figure 1E) to show that in particular in response to BTZ, the proteasome remains inhibited for 6 hrs, well beyond the turnover observed in Figure 1D. The question is: Does this response in eHAP cells reflect the situation in primary B-cells? Could this question of how long the proteasome remains inhibited be answered directly in primary B-cells?

It should be clarified which cells were used in Figure 1F. The authors write: "Thus, proteasome activity is not readily recovered even after the removal of proteasome inhibitor, which contrasts starkly with the rapid recovery of TOP2 enzymatic activity." What is the direct evidence for the rapid recovery of TOP2 enzymatic activity? The evidence of Figure 1E is indirect.

5) The results of Figure 5, that the repair of Etoposide-induced DNA damage is not repaired by NHEJ (using the Ligase IV mutant) or HR (using the BRCA1 mutant) leaves the obvious question: which DSB repair pathway is responsible for joining these DSBs? While an experimental answer would go well beyond what a revision should achieve, the authors should discuss this. The observed DSB resection at etoposide-induced DSBs would be consistent with repair by aNHEJ/MMEJ and may suggest a role of Polθ. How clear is it that the BRCA1 defect inhibits all DSB repair by HR? Is DSB resection at etoposide induced DSB normal in BRCA1-deficient cells?

6) Figure 6A lacks error bars. Was this a single experiment? Are the results reproducible? The data for N=3 and error bars should be included.

7) The authors suggest in the Discussion that this work implies that the timing of etoposide with proteasome inhibitors for cancer cells needs to take this work into consideration, but the authors do not formally show that this same effect occurs in tumor cells. As this appears to be a very clinically relevant and important hypothesis, I believe the work could be improved by showing in two tumor cell lines whether the timing of proteasome inhibition influences the extent of chromosome aberrations induced by etoposide. As the authors analyze cycling MEFs in this work, it is assumed that the proliferative nature of the tumor cells will not interfere with this experiment.

8) There is a lack of adequate references to prior work in this manuscript:

As examples:

a) The authors state: "It has been suggested that a robust DNA damage response (DDR) is elicited only after ETO-trapped TOP2ccs have been converted to "clean" protein-free DSBs (Zhang et al., 2006)". These publications should be cited as they show this is the case: Mårtensson et al., 2003, Muslimović et al., 2009 (PMCID: PMC2689654), Sunter et al., 2010 (PMCID: PMC2935090), Lee et al., 2018 (PMCID: PMC6073685).

b) The authors state: "Other studies have shown that inhibiting the proteasome not only preserves the reversibility of TOP2ccs but also is able to suppress DDR signaling (Mao et al., 2001); Zhang et al., 2006). However, the implications of suppressing the DDR in response to ETO using proteasome inhibitors, on both the long-term genome integrity and overall viability of the cell, have not been fully explored." The authors should look more carefully at, and cite where appropriate, other studies looking at combinations of proteasome inhibitors and drugs targeting TOP2:

Lee et al., 2016 (PMCID: PMC5071433), Metzler et al., 2009, Dittus et al., 2018, Aras and Yerlikaya, 2016 (PMCID: PMC4841005), Destanovic, Boos and Lanvers-Kaminsky, 2018.

c) The half-lives of TOP2 complexes following ETO removal has previously been reported for both TOP2A and TOP2B; TOP2B has a shorter half-life than TOP2A.

This should be addressed in the text: Willmore et al., 1998, Errington et al., 2004, Lee et al., 2016 (PMCID: PMC5071433).

d) The authors report that γ-H2AX signal is delayed by proteasomal inhibition preventing processing of TOP2 covalent complexes. This have been reported before by several groups: Mårtensson et al., 2003, Muslimović et al., 2009 (PMCID: PMC2689654), Sunter et al., 2010 (PMCID: PMC2935090), Lee et al., 2018 (PMCID: PMC6073685).

[Editors' note: further revisions were suggested prior to acceptance, as described below.]

Thank you for resubmitting your work entitled "Suppressing Proteasome Mediated Processing of Topoisomerase II DNA-Protein Complexes Preserves Genome Integrity" for further consideration by *eLife*. Your revised article has been evaluated by Maureen Murphy as the Senior and Reviewing Editor.

The manuscript has been improved and there are no new experiments required, but there are some remaining issues that need to be addressed within the text before final acceptance, as outlined below.

1) In some places you mention "spontaneous reversal" of complexes; a more precise way of stating this, would be "reversal of complexes by completion of the enzymes' catalytic cycle upon drug removal."

2) You clarify that there are two replicates of the END-seq data, with ~ 70% overlap between the two replicates (supplementary figure). Data for one gene are shown in Figures 2A and 3A and various types of data analysis on the END-seq data are shown. Were these data generated from just one replicate or are they combined data from the two replicates, and if the latter, was it just on the ~ 70% of the data that was concordant between the two replicas? Can you please clarify this?

3) END-seq is a novel approach in this study; it would be good to have a bit more discussion of the END-seq data. In Figure 3A resection data is shown; how was this determined/measured? Also can you comment as to what proportion of the breaks were SSBs versus DSBs? And how does this data compare with what has been recently reported by Gittens et al., 2019?

4) This study reports TOP2B complex levels following short term exposure to proteasome inhibitors prior to the TOP2 drug etoposide; this has been previously studied and published by Lee et al., 2016, for both TOP2A and TOP2B, using the TARDIS assay (Figure 9 in Lee et al., 2016). The current study and Lee et al. report similar findings, that the complexes are short lived and remain for longer when the proteasome is inhibited. Thus, the biochemical studies in both studies produce comparable results; this should be clearly mentioned.

---

## [Author Response]

1) According to the present draft, many of the experiments appear to have only been done once: for example, Figure 1B, C, D, F; Figure 2D and G; Figure 3B; Figure 4B and D; Figure 5B and C; Figure 6A. The authors need to more accurately state how many biological and technical replicates were done, and how exactly statistical analysis was done. Whereas a table stating such was provided to the reviewers, this information needs to be now included in clear manner in a revised manuscript.

As requested, we have added a table (Supplementary file 1) with the number and type of replicates, as well as an explanation of the statistical analyses for each of the figures.

2) Figure 1D is a slot blot of ICE assay fractions. It is unclear what the difference is between ETO and ETO + 0 hr washout; or between pre-MG132 + ETO and pre-MG132 + ETO + 0 hr washout. It seems this was done only once. Why? In the reference cited for the method it recommends doing the ICE assay N=3. In order to make quantitative conclusions from this type of data, the experiment must be carried out several times (at least three) so that statistical testing can be applied. And why has only TOP2B been probed for and not TOP2A? If this is because the cells used express very little TOP2A then this should be stated (and preferably demonstrated).

We have now performed the ICE assay three times (N=3) and updated Figure 1D to include new experimental data. The difference between ETO and ETO + 0 hr washout is that cells were harvested in the presence of ETO in the former, while in the latter, cells were harvested immediately after ETO was washed out three times in cold media and centrifuged for 5 minutes at 1500 rpm to pellet cells. Additional details are provided in the text, Materials and methods and figure legends. The same explanation applies to pre-MG132 + ETO and pre-MG132 + ETO + 0 hr washout. This has been clarified in the figure legends, Materials and methods and text.

To address the reviewers’ comments about the levels of TOP2B vs. TOP2A, we have updated the text to clarify that TOP2B is the primary isoform present in G1 primary mouse B-cells. We refer to a previous paper from our group where levels of TOP2A and TOP2B were measured in the same primary mouse B-cells (Canela et al., 2017) used in these experiments. We, therefore, did not perform a TOP2A ICE assay.

3) In several cases there are not enough details about methods, and it is requested that the authors carefully go through the manuscript to ensure there are enough details that will enable others to reproduce their findings:

We appreciate the reviewers’ inputs on these issues.

a) The section on DNA END-seq is novel, yet it is not clearly detailed enough for others to repeat.

We have improved and clarified the references in the Materials and methods section to indicate where readers can find a detailed END-seq protocol which was followed precisely (Canela et al., 2017; Canela et al., 2019).

b) In the text it says ETO was washed out "immediately", but in the Materials and methods section, it says cells were washed three times in drug free media. Washing three times would take time. The authors should state how long this took and at what temperature was the washing done. These details are necessary to interpret the reported data.

We have amended the text and Materials and methods to detail the washout procedure (3x5 min wash in cold media, 1500 rpm, 4°C).

c) Figure 3A and C – not clear images, and not clear how many replicates. Please clarify how long after the drug treatments were the mitotic spreads produced.

We have improved the image quality of the metaphase spreads (now in Figures 4A and C) and added color-coded arrowheads to highlight the different types of chromosomal aberrations. We have also updated the figure legend to indicate the timepoint post-ETO treatment (24 hr) at which the mitotic spreads were processed.

4) In Figure 1, the authors use primary mouse B-cells to show that pre-treatment with proteasome inhibitors suppresses the γ-H2AX response to Etoposide. After washout of the drugs, TOP2 trapped complexes quickly turn over, and the authors suspect that this is due to resumption of TOP2 activity rather than reactivation of the proteasome. To support this conclusion, the authors conduct an experiment in eHAP cells (Figure 1E) to show that in particular in response to BTZ, the proteasome remains inhibited for 6 hrs, well beyond the turnover observed in Figure. 1D. The question is: Does this response in eHAP cells reflect the situation in primary B-cells? Could this question of how long the proteasome remains inhibited be answered directly in primary B-cells?It should be clarified which cells were used in Figure 1F. The authors write: "Thus, proteasome activity is not readily recovered even after the removal of proteasome inhibitor, which contrasts starkly with the rapid recovery of TOP2 enzymatic activity." What is the direct evidence for the rapid recovery of TOP2 enzymatic activity? The evidence of Figure 1E is indirect.

To directly address the reviewers’ concerns about the longevity of proteasomal inhibition in primary B-cells, we assessed pan-p53 protein levels by western blot at several timepoints following proteasome inhibitor treatment. Proteasome inhibition has been reproducibly shown to cause accumulation of p53 (An et al., 2000 (PMID:10914553); Halasi, Pandit and Gartel, 2014 (PMID: 25485499)). Indeed, we found that both BTZ and MG132 treatment led to accumulation of p53 in primary B cells. Moreover, elevated p53 protein levels persisted for many hours after proteasome inhibitors were removed. The kinetics of p53 accumulation in B cells was comparable to YFP accumulation in eHAP cells suggesting that proteasome activity remains suppressed for an extended time after the wash out of ETO and proteasome inhibitors. We have now included a representative pan-p53 Western blot in the revised Figure 1 (now Figure 1F). We also observed similar results when we measured the total level of ubiquitinated proteins over a similar time course (data not shown). We have revised the text accordingly.

The reviewer makes a good point and is entirely correct that Figure 1D and E only provided indirect evidence for the rapid recovery of TOP2 enzymatic activity. We have, therefore, tempered our conclusion by stating: “Thus, proteasome activity is not readily recovered even after the removal of proteasome inhibitor, suggesting that the rapid loss of ETO-induced TOP2ccs in MG132 pre-treated cells upon washout most likely reflected spontaneous recovery of TOP2 enzymatic activity.”

5) The results of Figure 5, that the repair of Etoposide-induced DNA damage is not repaired by NHEJ (using the Ligase IV mutant) or HR (using the BRCA1 mutant) leaves the obvious question: which DSB repair pathway is responsible for joining these DSBs? While an experimental answer would go well beyond what a revision should achieve, the authors should discuss this. The observed DSB resection at etoposide-induced DSBs would be consistent with repair by aNHEJ/MMEJ and may suggest a role of Polθ. How clear is it that the BRCA1 defect inhibits all DSB repair by HR? Is DSB resection at etoposide induced DSB normal in BRCA1-deficient cells?

We thank the reviewers for bring up this important question and have tried to address it experimentally. Like the reviewers, we were also confounded by the fact neither classical NHEJ nor HR appeared to be absolutely critical for mediating error-prone repair of ETO-induced DSBs. To further address this question, as well as concerns that BRCA1 function may not be totally essential for HR, we have performed additional experiments in B cells lacking BRCA2 or Polθ. The new results are now shown in Figure 5 and Figure 5—figure supplement 1. Briefly, we observed that loss of BRCA2 or Polθ did not significantly attenuate chromosomal rearrangements in ETO-treated B cells. These data strengthened our initial conclusion that no single repair pathway is essential to produce genome instability by ETO. Thus, the large number of DSBs induced by high dose ETO likely engages multiple repair pathways, all of which carry some potential for misrepair. We speculate that it may be possible to suppress ETO-induced genome instability by inhibiting multiple DSB repair pathways at once. However, this would be a major undertaking.

6) Figure 6A lacks error bars. Was this a single experiment? Are the results reproducible? The data for N=3 and error bars should be included.

We have now performed the western blot for γ-H2AX in WT vs. RNF4^-/-^ MEFs three times (N=3) and updated Figure 6A to include both a western blot as well as a bar graph with error bars. Statistical information is noted in the figure legend.

7) The authors suggest in the Discussion that this work implies that the timing of etoposide with proteasome inhibitors for cancer cells needs to take this work into consideration, but the authors do not formally show that this same effect occurs in tumor cells. As this appears to be a very clinically relevant and important hypothesis, I believe the work could be improved by showing in two tumor cell lines whether the timing of proteasome inhibition influences the extent of chromosome aberrations induced by etoposide. As the authors analyze cycling MEFs in this work, it is assumed that the proliferative nature of the tumor cells will not interfere with this experiment.

We thank the reviewers for this suggestion. To test this, we have performed CellTiter-Glo viability assays in two human tumor cell lines (DMS114 a small cell lung cancer cell line, and HeLa cells). The cells were pretreated or not with BTZ and then exposed to graded doses of ETO for 1 hour. There was considerable variability among the cancer cell lines in terms of their individual sensitivities to ETO and BTZ. Nevertheless, taken in total, our results showed that a 1 hr BTZ pre-treatment was able to modestly increase the viability in both ETO-treated human cancer cells (see Author response image 1). Although we note that the cytoprotective effects of proteasome inhibition were not as dramatic as what was observed in primary B-cells, the protective trend of proteasome inhibition held true, and reinforced our argument that caution should be exercised when designing *treatment* regimens that combine TOP2 poisons and proteasome inhibitors.

8) There is a lack of adequate references to prior work in this manuscript:As examples:a) The authors state: "It has been suggested that a robust DNA damage response (DDR) is elicited only after ETO-trapped TOP2ccs have been converted to "clean" protein-free DSBs (Zhang et al., 2006)". These publications should be cited as they show this is the case: Mårtensson et al., 2003, Muslimović et al., 2009 (PMCID: PMC2689654), Sunter et al., 2010 (PMCID: PMC2935090), Lee et al., 2018 (PMCID: PMC6073685).b) The authors state: "Other studies have shown that inhibiting the proteasome not only preserves the reversibility of TOP2ccs but also is able to suppress DDR signaling (Mao et al., 2001); Zhang et al., 2006). However, the implications of suppressing the DDR in response to ETO using proteasome inhibitors, on both the long-term genome integrity and overall viability of the cell, have not been fully explored." The authors should look more carefully at, and cite where appropriate, other studies looking at combinations of proteasome inhibitors and drugs targeting TOP2:Lee et al., 2016 (PMCID: PMC5071433), Metzler et al., 2009, Dittus et al., 2018, Aras and Yerlikaya, 2016 (PMCID: PMC4841005), Destanovic, Boos and Lanvers-Kaminsky, 2018.c) The half-lives of TOP2 complexes following ETO removal has previously been reported for both TOP2A and TOP2B; TOP2B has a shorter half-life than TOP2A. This should be addressed in the text: Willmore et al., 1998, Errington et al., 2004, Lee et al., 2016 (PMCID: PMC5071433).d) The authors report that γ-H2AX signal is delayed by proteasomal inhibition preventing processing of TOP2 covalent complexes. This have been reported before by several groups: Mårtensson et al., 2003, Muslimović et al., 2009 (PMCID: PMC2689654), Sunter et al., 2010 (PMCID: PMC2935090), Lee et al., 2018 (PMCID: PMC6073685).

We thank the reviewers for directing us to references that should have been cited in the original submission. We have now added these references where appropriate in the manuscript.

[Editors' note: further revisions were suggested prior to acceptance, as described below.]

[…]1) In some places you mention "spontaneous reversal" of complexes; a more precise way of stating this, would be "reversal of complexes by completion of the enzymes' catalytic cycle upon drug removal."

As requested, we have replaced the phrase “spontaneous reversal” with the phrase “reversal of complexes by completion of the enzymes' catalytic cycle upon drug removal” or “reversal of TOP2ccs by completion of the enzymes' catalytic cycle upon drug removal”, where appropriate in the text.

2) You clarify that there are two replicates of the END-seq data, with ~ 70% overlap between the two replicates (supplementary figure). Data for one gene are shown in Figures 2A and 3A and various types of data analysis on the END-seq data are shown. Were these data generated from just one replicate or are they combined data from the two replicates, and if the latter, was it just on the ~ 70% of the data that was concordant between the two replicas? Can you please clarify this?

As requested, we have clarified how the END-seq data analysis was performed in the Materials and methods section. Briefly, as shown in Figure 2—figure supplement 1, the END-seq signal at ETO break sites is very reproducible between replicates. The data presented in Figures 2 and 3 are generated from one of the END-seq replicates. We have performed the same analyses for the other replicate, and the results are consistent. We have updated Figure 2—figure supplement 1 with some further analysis (Figure 2—figure supplement 1C-F) using data from second replicate to support the results shown in Figures 2B-E to illustrate this point and added Figure 3—figure supplement 1A-C to support the results from Figure 3B-D).

3) END-seq is a novel approach in this study; it would be good to have a bit more discussion of the END-seq data. In Figure 3A resection data is shown; how was this determined/measured? Also can you comment as to what proportion of the breaks were SSBs versus DSBs? And how does this data compare with what has been recently reported by Gittens et al., 2019?

We thank the reviewers for their comments, while ETO can generate high level of SSBs as detailed in previous studies and in Glittens et al., END-seq only monitors DSBs that are generated by ETO, which is also mentioned in Glittens et al. Therefore, we have updated the text in the Results section, when discussing the use END-seq, to clarify this point and cite Glittens et al. to support it. In regard to the methodology to quantify resection, there is a section in the Materials and methods labelled “Resection Quantification” that details how the analysis was performed.

4) This study reports TOP2B complex levels following short term exposure to proteasome inhibitors prior to the TOP2 drug etoposide; this has been previously studied and published by Lee et al., 2016, for both TOP2A and TOP2B, using the TARDIS assay (Figure 9 in Lee et al., 2016). The current study and Lee et al. report similar findings, that the complexes are short lived and remain for longer when the proteasome is inhibited. Thus, the biochemical studies in both studies produce comparable results; this should be clearly mentioned.

We thank the reviewers for clarifying that this reference should have been cited more explicitly in the previous submission. We have now updated the text to reflect the findings from this reference with regard to our results measuring TOP2B complex levels by ICE assay.